# UNICORN: A UNIFIED BACKDOOR TRIGGER INVERSION FRAMEWORK

**Zhenting Wang, Kai Mei, Juan Zhai, Shiqing Ma**
Department of Computer Science, Rutgers University
`{zhenting.wang,kai.mei,juan.zhai,sm2283}@rutgers.edu`

## ABSTRACT

The backdoor attack, where the adversary uses inputs stamped with triggers (e.g., a patch) to activate pre-planted malicious behaviors, is a severe threat to Deep Neural Network (DNN) models. Trigger inversion is an effective way of identifying backdoor models and understanding embedded adversarial behaviors. A challenge of trigger inversion is that there are many ways of constructing the trigger. Existing methods cannot generalize to various types of triggers by making certain assumptions or attack-specific constraints. The fundamental reason is that existing work does not consider the trigger's design space in their formulation of the inversion problem. This work formally defines and analyzes the triggers injected in different spaces and the inversion problem. Then, it proposes a unified framework to invert backdoor triggers based on the formalization of triggers and the identified inner behaviors of backdoor models from our analysis. Our prototype UNICORN is general and effective in inverting backdoor triggers in DNNs. The code can be found at https://github.com/RU-System-Software-and-Security/UNICORN.

## 1 INTRODUCTION

Backdoor attacks against Deep Neural Networks (DNN) refer to the attack where the adversary creates a malicious DNN that behaves as expected on clean inputs but predicts a predefined target label when the input is stamped with a trigger (Liu et al., 2018b; Gu et al., 2017; Chen et al., 2017; Liu et al., 2019; Wang et al., 2022c; Barni et al., 2019; Nguyen & Tran, 2021). The malicious models can be generated by data poisoning (Gu et al., 2017; Chen et al., 2017) or supply-chain attacks (Liu et al., 2018b; Nguyen & Tran, 2021). The adversary can choose desired target label(s) and the trigger. Existing work demonstrates that DNNs are vulnerable to various types of triggers. For example, the trigger can be a colored patch (Gu et al., 2017), a image filter (Liu et al., 2019) and a warping effect (Nguyen & Tran, 2021). Such attacks pose a severe threat to DNN based applications especially those in security-critical tasks such as malware classification (Severi et al., 2021; Yang et al., 2022; Li et al., 2021a), face recognition (Sarkar et al., 2020; Wenger et al., 2021), speaker verification (Zhai et al., 2021), medical image analysis (Feng et al., 2022), brain-computer interfaces (Meng et al., 2020), and autonomous driving (Gu et al., 2017; Xiang et al., 2021).

Due to the threat of backdoor attacks, many countermeasures have been proposed. For example, anti-poisoning training (Li et al., 2021c; Wang et al., 2022a; Hong et al., 2020; Tran et al., 2018; Hayase et al., 2021; Chen et al., 2018) and runtime malicious inputs detection (Gao et al., 2019; cho; Doan et al., 2020; Zeng et al., 2021). Different from many methods that can only work under specific threat models (e.g., anti-poisoning training can only work under the data-poisoning scenario), trigger inversion (Wang et al., 2019; Liu et al., 2019; Guo et al., 2020; Chen et al., 2019; Shen et al., 2021) is practical and general because it can be applied in both poisoning and supply-chain attack scenarios. It is a post-training method where the defender aims to detect whether the given model contains backdoors. It reconstructs backdoor triggers injected in the model as well as the target labels, which helps analyze the backdoors. If there exists an inverted pattern that can control the predictions of the model, then it determines the model is backdoored. Most existing trigger inversion methods (Guo et al., 2020; Liu et al., 2019; Shen et al., 2021; Chen et al., 2019) are built on Neural Cleanse (Wang et al., 2019), which assumes that the backdoor triggers are static patterns in the pixel space. It defines a backdoor sample as $\tilde{x} = (1 - m) \odot x + m \odot t$, where $m$ and $t$ are pixel space trigger mask

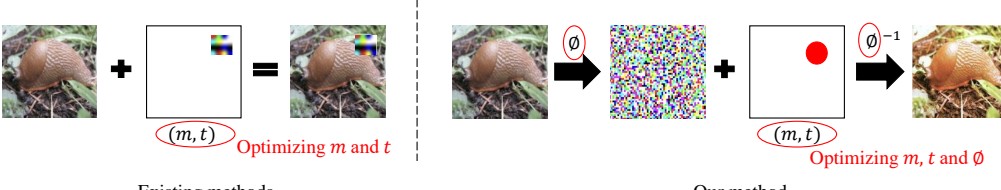

Fig. 1: Existing trigger inversion methods and our method.

and trigger pattern. It inverts the trigger via devising an optimization problem that searches for a small and static pixel space pattern. Such methods achieve good performance on the specific type of triggers that are static patterns in the pixel space (Gu et al., 2017; Chen et al., 2017). However, it can not generalize to different types of triggers such as image filters (Liu et al., 2019) and warping effects (Nguyen & Tran, 2021). Existing methods fail to invert various types of triggers because they do not consider the trigger's design space in their formulation of the inversion problem.

In this paper, we propose a trigger inversion framework that can generalize to different types of triggers. We first define a backdoor trigger as a predefined perturbation in a particular input space. A backdoor sample $\tilde{x}$ is formalized as $\tilde{x} = \phi^{-1}\left((1 - m) \odot \phi(x) + m \odot t\right)$, where $m$ and $t$ are input space trigger mask and trigger pattern, $x$ is a benign sample, $\phi$ is an invertible input space transformation function that maps from the pixel space to other input spaces. $\phi^{-1}$ is the inverse function of $\phi$, i.e., $x = \phi^{-1}\left(\phi(x)\right)$. As shown in Fig. 1, the critical difference between our framework and existing methods is that we introduce an input space transformation $\phi$ to unify the backdoor triggers injected in different spaces. Besides searching for the pixel space trigger mask $m$ and pattern $t$ as existing methods do, our method also searches the input space transformation function $\phi$ to find the input space where the backdoor is injected. We also observe successful backdoor attacks will lead to compromised activation vectors in the intermediate representations when the model recognizes backdoor triggers. Besides, we find that the benign activation vector will not affect the predictions of the model when the compromised activation values are activated. This observation means the compromised activation vector and the benign activation vector are disentangled in successful backdoor attacks. Based on the formalization of triggers and the observation of the inner behaviors of backdoor models, we formalize the trigger inversion as a constrained optimization problem.

Based on the devised optimization problem, we implemented a prototype UNICORN (**Uni**fied Ba**c**kd**o**or T**r**igger I**n**version) in PyTorch and evaluated it on nine different models and eight different backdoor attacks (i.e., Patch attack (Gu et al., 2017), Blend attack (Chen et al., 2017), SIG (Barni et al., 2019), moon filter, kelvin filter, 1977 filter (Liu et al., 2019), WaNet (Nguyen & Tran, 2021) and BppAttack (Wang et al., 2022c)) on CIFAR-10 and ImageNet dataset. Results show UNICORN is effective for inverting various types of backdoor triggers. On average, the attack success rate of the inverted triggers is 95.60%, outperforming existing trigger inversion methods.

Our contributions are summarized as follows: We formally define the trigger in backdoor attacks. Our definition can generalize to different types of triggers. We also find the compromised activations and the benign activations in the model's intermediate representations are disentangled. Based on the formalization of the backdoor trigger and the finding of intermediate representations, we formalize our framework as a constrained optimization problem and propose a new trigger inversion framework. We evaluate our framework on nine different DNN models and eight backdoor attacks. Results show that our framework is more general and more effective than existing methods. Our open-source code can be found at https://github.com/RU-System-Software-and-Security/UNICORN.

## 2 BACKGROUND & MOTIVATION

**Backdoor.** Existing works (Turner et al., 2019; Salem et al., 2022; Nguyen & Tran, 2020; Tang et al., 2021; Liu et al., 2020; Lin et al., 2020; Li et al., 2020; Chen et al., 2021; Li et al., 2021d; Doan et al., 2021b; Tao et al., 2022d; Bagdasaryan & Shmatikov, 2022; Qi et al., 2023; Chen et al., 2023) demonstrate that deep neural networks are vulnerable to backdoor attacks. Models infected with backdoors behave as expected on normal inputs but present malicious behaviors (i.e., predicting a certain label) when the input contains the backdoor trigger. Existing methods defend against backdoor attacks during training (Du et al., 2020; Hong et al., 2020; Huang et al., 2022; Li et al., 2021c; Wang et al., 2022a; Hayase et al., 2021; Tran et al., 2018; Zhang et al., 2023), or detect

malicious inputs during inference (Gao et al., 2019; Doan et al., 2020; cho; Zeng et al., 2021; Guo et al., 2023), or remove and mitigate backdoors in the given model offline (Liu et al., 2018a; Li et al., 2021b; Zeng et al., 2022; Wu & Wang, 2021; Tao et al., 2022b; Cheng et al., 2023).

**Trigger Inversion.** Trigger inversion (Wang et al., 2019; Liu et al., 2019; Guo et al., 2020; Chen et al., 2019; Wang et al., 2022b; Liu et al., 2022b; Tao et al., 2022c; Liu et al., 2022a; Shen et al., 2022) is a post-training approach to defending against backdoor attacks. Compared to training-time backdoor defenses, this method can also defend against supply-chain backdoor attacks. Meanwhile, this approach is more efficient than inference-time defenses. Moreover, it can recover the used trigger, providing more information and have many applications such as filtering out backdoor samples and mitigating the backdoor injected in the models. It achieves good results in many existing research papers (e.g., Wang et al. (2019), Liu et al. (2019), Guo et al. (2020), Shen et al. (2021), Hu et al. (2022)) and competitions (e.g., NIST TrojAI Competition (tro)), showing it is a promising direction. The overreaching idea is to optimize a satisfactory trigger that can fool the model. Existing trigger inversion methods achieve promising performance on specific triggers. However, they are not generalizable by making certain assumptions or constraints during optimizing the trigger. During the optimization, it assumes that the trigger is a static pattern with a small size in the pixel space. Such assumption is suitable for pixel space attacks (e.g., BadNets (Gu et al., 2017) and Blend attack (Chen et al., 2017)) but does not hold for feature-space attacks that have dynamic pixel space perturbations, e.g., WaNet (Nguyen & Tran, 2021). Most existing approaches (Shen et al., 2021; Liu et al., 2019; Guo et al., 2020; Chen et al., 2019; Hu et al., 2022) follow the similar assumption and thus have the same limitation. In this paper, we propose a general trigger inversion framework.

## 3 BACKDOOR ANALYSIS

### 3.1 THREAT MODEL

**Attacker's Goal.** Backdoor attacks aim to generate a backdoor model $\mathcal{M}$ s.t. $\mathcal{M}(\boldsymbol{x}) = y, \mathcal{M}(\tilde{\boldsymbol{x}}) = y_t$, where $\boldsymbol{x}$ is a clean sample, $\tilde{\boldsymbol{x}}$ is a backdoor sample (with the trigger), and $y_t \neq y$ is the target label. A successful backdoor attack achieves the following goals:

*Effectiveness.* The backdoor model shall have a high attack success rate on backdoor inputs while maintaining high accuracy on benign inputs.

*Stealthiness.* The backdoor trigger shall not change the ground truth label of the input, i.e., a benign input and its malicious version (with trigger) shall be visually similar.

**Defender's Goal & Capability.** In this paper, we focus on reconstructing the backdoor triggers injected into the infected models. Following existing trigger inversion methods (Wang et al., 2019; Liu et al., 2019; Guo et al., 2020; Chen et al., 2019), we assume a small dataset containing correctly labeled benign samples is available and defenders have access to the target model. Note that our trigger inversion method does not require knowing the target label, it conducts trigger inversion for all labels and identifies the potential target label.

### 3.2 FORMALIZING BACKDOOR TRIGGERS

In software security, backdoor attack mixes malicious code into benign code to hide malicious behaviors or secrete access to the victim system, which are activated by trigger inputs. Backdoor attacks in DNN systems share the same characteristics. Trigger inversion essentially recovers the backdoor trigger by finding input patterns that activate such behaviors. Existing trigger inversion methods (Wang et al., 2019; Liu et al., 2019; Guo et al., 2020; Shen et al., 2021) can not generalize to different types of triggers. They define the backdoor samples as $\tilde{\boldsymbol{x}} = (1 - \boldsymbol{m}) \odot \boldsymbol{x} + \boldsymbol{m} \odot \boldsymbol{t}$, where $\boldsymbol{m}$ and $\boldsymbol{t}$ are pixel space trigger mask and trigger pattern. The reason why they can not invert different types of triggers is that existing attacks inject triggers in other input spaces (Liu et al., 2019; Wang et al., 2022c; Nguyen & Tran, 2021) while existing trigger inversion methods cannot handle them all. In this paper, we define backdoor triggers as:

**Definition 1** *(Backdoor Trigger) A backdoor trigger is a mask $\boldsymbol{m}$ and content pattern $\boldsymbol{t}$ pair $(\boldsymbol{m}, \boldsymbol{t})$ so that for a pair of functions $\phi$ and $\phi^{-1}$ that transfer an image from pixel space to other input spaces that satisfy $\boldsymbol{x} = \phi^{-1}(\phi(\boldsymbol{x}))$ for an input $\boldsymbol{x}$ represented in the pixel space, we have a backdoor sample $\tilde{\boldsymbol{x}} = \phi^{-1}((1 - \boldsymbol{m}) \odot \phi(\boldsymbol{x}) + \boldsymbol{m} \odot \boldsymbol{t})$.*

| Space | Pixel | | | Signal | | | Feature | Numerical |
|---|---|---|---|---|---|---|---|---|
| Attack | Patch | Blend | SIG | 1977 | Kelvin | Moon | WaNet | BppAttack |
| Backdoor Samples | | | | | | | | |

Fig. 2: Backdoor attacks in different spaces.

The difference between our definition and that of existing works is that we introduce an input space transformation function pair $\phi$ and $\phi^{-1}$ that convert images from/to the pixel space to/from other spaces. The input space transformation function $\phi$ is invertible, i.e., $\boldsymbol{x} = \phi^{-1}(\phi(\boldsymbol{x}))$. For pixel space attacks, $\phi^{-1}(\boldsymbol{x}) = \phi(\boldsymbol{x}) = \boldsymbol{x}$. Fig. 2 classifies input spaces used by existing attacks. **Pixel Space Attacks** mixes the malicious pixels and benign contents at the pixel level. For example, patch attack (Gu et al., 2017) directly uses a static colored patch as a trigger; Blend attack (Chen et al., 2017) generates backdoor samples via blending the images with a predefined pattern; SIG attack (Barni et al., 2019) uses the sinusoidal signal pattern to create backdoor samples. In **Signal Space Attacks**, the adversary mixes the malicious signals with benign ones. For instance, the Filter attack (Liu et al., 2019) uses an image signal processing kernel (e.g., 1977, Kelvin and Moon filters used in Instagram) to generate backdoor samples. For **Feature Space Attacks**, backdoor samples inject malicious abstracted features and benign ones. As an example, WaNet (Nguyen & Tran, 2021) introduces the warping feature as a trigger. To perform **Numerical Space Attacks**, the attacker creates backdoor samples by changing numerical representations, e.g., the BppAttack (Wang et al., 2022c) generates backdoor samples via introducing quantization effects into the numerical representation of images. Existing trigger inversion methods (Wang et al., 2019; Liu et al., 2019; Guo et al., 2020; Shen et al., 2021) define backdoor triggers in the pixel space, and can not generalize to the attacks in different spaces. Compared to existing works, our definition is more general, and it can represent the triggers in different spaces.

### 3.3 BACKDOOR BEHAVIORS IN INTERMEDIATE REPRESENTATIONS

Directly inverting the trigger can not guarantee that the inverted transformations have trigger effects. It will yield general transformations which are not specific to the injected backdoor. Thus, we need to find the invariant in the backdoor behavior and use it to constrain the inverted trigger. A successful backdoor attack will lead to compromised activation values that are highly associated with backdoor triggers. The intermediate representation of DNNs contains the activation values of different neurons, the composite of these values can be viewed as a vector. In this section, we use $\boldsymbol{u}$ to represent the unit vector in the intermediate representations. $\boldsymbol{A}$ is the activation values in the intermediate representations. We also use $\boldsymbol{u}(\boldsymbol{x})$ to denote the projection length on unit vector $\boldsymbol{u}$ for input $\boldsymbol{x}$'s intermediate representation. Their formal definition can be found in §A.1.

In backdoored DNNs, the intermediate activation vector $\boldsymbol{A}$ can be decomposed into compromised activation vector $\boldsymbol{A}_c$ and the benign activation vector $\boldsymbol{A}_b$, i.e., $\boldsymbol{A} = \boldsymbol{A}_c + \boldsymbol{A}_b$. $\boldsymbol{A}_c$ is on direction of unit vector $\boldsymbol{u}_c$ (compromised direction) that is most sensitive to the backdoor triggers:

$$\boldsymbol{u}_c = \max_{\boldsymbol{u}} \|\boldsymbol{u}(\tilde{\boldsymbol{x}}) - \boldsymbol{u}(\boldsymbol{x})\|, \boldsymbol{x} \in \mathcal{X} \tag{1}$$

where $\tilde{\boldsymbol{x}}$ is the samples pasted with trigger and $\boldsymbol{x}$ is the clean sample. Meanwhile, $\boldsymbol{A}_b$ is on direction of unit vector $\boldsymbol{u}_b$ that is not sensitive to the backdoor triggers (i.e., $\|\boldsymbol{u}_c(\tilde{\boldsymbol{x}}) - \boldsymbol{u}_c(\boldsymbol{x})\| < \tau$, where $\tau$ is a threshold value), and sensitive to the benign contents. As shown in Eq. 2, given a backdoor model, the backdoor trigger will lead to a specific projection value on compromised direction $\boldsymbol{u}_c$, where $F$ is the trigger pasting function to create the backdoor sample $\tilde{\boldsymbol{x}}$ via benign sample $\boldsymbol{x}$, i.e., $\tilde{\boldsymbol{x}} = F(\boldsymbol{x})$. $\boldsymbol{S}$ is the backdoor vector.

$$\boldsymbol{x} \in F(\mathcal{X}) \implies \boldsymbol{A}_c = \boldsymbol{u}_c(\boldsymbol{x}) \approx \boldsymbol{S} \tag{2}$$

**Corollary 1** *When the compromised activation vector $\boldsymbol{A}_c$ is close to the backdoor vector $\boldsymbol{S}$, the model $\mathcal{M}$ will predict the target label $y_t$ regardless of benign activation vector $\boldsymbol{A}_b$, i.e., $\forall \boldsymbol{A}_b, \boldsymbol{A}_c \approx \boldsymbol{S} \implies g(\boldsymbol{A}_c, \boldsymbol{A}_b) = y_t$, where $g$ is the sub-model from the intermediate layer to the output layer.*

As discussed in §3.1, a backdoor attack is effective means it has a high attack success rate and high benign accuracy simultaneously. If the model has high benign accuracy, then it can extract the benign features well. Namely, different benign features will cause different activation values in

benign activation vector. Meanwhile, if the attack success rate is high, then the model will predict target labels if it recognizes the trigger, regardless of the benign contents. Also, backdoor trigger will cause specific intermediate representations in model. Based on the above analysis, we conclude that the backdoor intermediate representations on the compromised activations will lead to the model predicting the target label while ignoring the activation values on the benign activation vector. More support for Corollary 1 can be found in §A.2. Existing works also find some patterns of backdoor behaviors, such as having a shortcut path when inferring (Zheng et al., 2021; Li et al., 2022). However, it is hard to integrated them with the optimization for trigger inversion (See §A.3).

## 4 OPTIMIZATION PROBLEM

Based on the discussed attack goals and observation, we propose a backdoor trigger inversion framework to reconstruct the injected triggers in the given models. In Definition 1, we introduce input space transformation functions $\phi$ and $\phi^{-1}$ to specify the input space where the backdoor is injected. Trigger inversion is an optimization problem finding the input space mask and pattern $\boldsymbol{m}$ and $\boldsymbol{t}$ as well as the input space transformation functions. Directly optimizing functions is challenging. Because of the high expressiveness of neural networks (Hornik et al., 1989), in our optimization problem, we use two neural networks $P$ and $Q$ to correspondingly approximate $\phi$ and its inverse function $\phi^{-1}$. In this section, we first discuss the objectives and the constraints for the optimization problem. Then we introduce the formalization and the implementation of our trigger inversion.

**Objectives.** Based on the goal of the attacker, the attack success rate (ASR) should be high. We achieve the effectiveness objective by optimizing the classification loss of inverted backdoor samples on the target label. Minimizing $\mathcal{L}\left(\mathcal{M}(\tilde{\boldsymbol{x}}), y_t\right)$ means optimizing the ASR for the inverted trigger. Here, $\mathcal{M}$ is the target model, $\tilde{\boldsymbol{x}}$ is the sample pasted with inverted trigger. Function $\mathcal{L}$ is the loss function of the model. $y_t$ is the target label.

**Invertible Constraint.** As $P$ and $Q$ are used to model the input space transformation function $\phi$ and its inverse function $\phi^{-1}$, we add the constraint that the inputs generated by composite function $Q \circ P$ should be close to the original inputs, i.e., $\|Q(P(\boldsymbol{x})) - \boldsymbol{x}\| < \alpha$, where $\alpha$ is a threshold value.

**Mask Size Constraint.** As pointed out in Neural Cleanse (Wang et al., 2019), the trigger can only modify a small part of the input space. Similar to Neural Cleanse, we constrain the size of the mask $\boldsymbol{m}$ in the input space introduced by transformation function $\phi$, i.e., $\|\boldsymbol{m}\| < \beta$.

**Stealthiness Constraint.** Based on the attacker's goal, the backdoor sample and the clean sample should be similar to the human visual system. Following existing works (Tao et al., 2022a; Cheng et al., 2021), we use SSIM score (Wang et al., 2004) to measure the visual difference between two images. Thus, the SSIM score between original sample $\boldsymbol{x}$ and the sample pasted with inverted trigger $\tilde{\boldsymbol{x}}$ should be higher than a threshold $\gamma$, i.e., $\text{SSIM}(\tilde{\boldsymbol{x}}, \boldsymbol{x}) > \gamma$.

**Disentanglement Constraint.** Based on our observation, the inverted trigger should lead to a specific projection value on the compromised activation vector. Meanwhile, the model will predict target labels regardless of the benign activations. We denote the constraint as disentangled constraint, i.e., $\boldsymbol{A}_c \perp \boldsymbol{A}_b$. To implement it, we devise a loss item $\mathcal{L}_{dis}$. Formally, it is defined in Eq. 3, where function $h$ and $g$ are the sub-model from the input layer to the intermediate layer and the sub-model from the intermediate layer to the output layer. By default, we separate $h$ and $g$ at the last convolutional layer in the model as it has well-abstracted feature representations. In Eq. 3, $\boldsymbol{m}'$ is the model's intermediate representation mask indicating the compromised activation direction. We constrain the size of $\boldsymbol{m}'$ (10% of the whole intermediate representation space by default) and use gradient descent to search the direction of the compromised activation vector. We then combine the compromised activation vector and benign activation vector. Given a input sample $\boldsymbol{x}$, we first compute the $\boldsymbol{A}_c$ via using the inverted trigger function $F$ (i.e., $F(\boldsymbol{x}) = \tilde{\boldsymbol{x}} = Q\left((1 - \boldsymbol{m}) \odot P(\boldsymbol{x}) + \boldsymbol{m} \odot \boldsymbol{t}\right)$). We then randomly select a set of different input samples $\boldsymbol{x}'$ and calculate the benign activation vector on it, i.e., $\boldsymbol{A}_b$. If the disentangle loss $\mathcal{L}_{dis}$ achieves low values, it means the benign activations can not influence the model's prediction when it recognizes backdoor representations on compromised activations. We consider it satisfy the disentanglement constraint is the value of $\mathcal{L}\left(g(\boldsymbol{A}_c, \boldsymbol{A}_b), y_t\right)$ is lower than a threshold value $\delta$.

$$
\begin{aligned}
&\mathcal{L}_{dis} = \mathcal{L}\left(g(\boldsymbol{A}_c, \boldsymbol{A}_b), y_t\right) + \|\boldsymbol{m}'\| \\
&\text{where } \boldsymbol{A}_c = \boldsymbol{m}' \odot h(F(\boldsymbol{x})), \ \boldsymbol{A}_b = (1 - \boldsymbol{m}') \odot h(\boldsymbol{x}'), \ \boldsymbol{x}' \neq \boldsymbol{x}
\end{aligned} \tag{3}
$$

**Formalization.**    Based on the objectives and constraints. We define our trigger inversion as a constrained optimization problem in Eq. 4, where $P$, $Q$, $\boldsymbol{m}$, $\boldsymbol{t}$, and $\boldsymbol{m}'$ are approximated input space transformation function, approximated inverse transformation function, backdoor mask and pattern in the infected space, intermediate representation mask, respectively. Function $\mathcal{L}$ is the loss function of the target neural network, $y_t$ is the target label, $P_\theta$ and $Q_\theta$ are the parameters of model $P$ and $Q$.

$$\min_{P_\theta, Q_\theta, \boldsymbol{m}, \boldsymbol{t}, \boldsymbol{m}'} \mathcal{L}\left(\mathcal{M}(\tilde{\boldsymbol{x}}), y_t\right)$$
$$\text{where } \tilde{\boldsymbol{x}} = Q\left((1 - \boldsymbol{m}) \odot P(\boldsymbol{x}) + \boldsymbol{m} \odot \boldsymbol{t}\right), \ \boldsymbol{x} \in \mathcal{X} \tag{4}$$
$$s.t. \|Q(P(\boldsymbol{x})) - \boldsymbol{x}\| < \alpha, \ \|\boldsymbol{m}\| < \beta, \ \text{SSIM}(\tilde{\boldsymbol{x}}, \boldsymbol{x}) > \gamma, \ \boldsymbol{A}_c \perp \boldsymbol{A}_b$$
$$\text{where } \boldsymbol{A}_c = \boldsymbol{m}' \odot h(\tilde{\boldsymbol{x}}), \ \boldsymbol{A}_b = (1 - \boldsymbol{m}') \odot h(\boldsymbol{x})$$

**Implementation.**    To solve the optimization problem formalized in Eq. 4, we also design a loss in Eq. 5. $w_1$, $w_2$, $w_3$ and $w_4$ are coefficient values for different items. Following existing methods (Wang et al., 2019; Liu et al., 2022b), we adjust these coefficients dynamically to satisfy the constraints. A detailed adjustment method can be found in the §A.6. We use a representative deep neural network UNet (Ronneberger et al., 2015) to model the space transformation function $\phi$. In detail, $P$ and $Q$ are represented by two identical UNet networks. By default, we set $\alpha = 0.01$, $\beta$ as 10% of the input space, $\gamma = 0.85$, and $\delta = 0.5$.

$$\mathcal{L}_{inv} = \mathcal{L}\left(\mathcal{M}(\tilde{\boldsymbol{x}}), y_t\right) + w_1 \cdot \|Q(P(\boldsymbol{x})) - \boldsymbol{x}\| + w_2 \cdot \|\boldsymbol{m}\| - w_3 \cdot \text{SSIM}(\tilde{\boldsymbol{x}}, \boldsymbol{x}) + w_4 \cdot \mathcal{L}_{dis} \tag{5}$$

## 5 EXPERIMENTS AND RESULTS

In this section, we first introduce the experiment setup (§5.1). We then evaluate the effectiveness of our method (§5.2) and conduct ablation studies (§5.3). We also evaluate UNICORN's generalizability to self-supervised models (§5.4).

### 5.1 EXPERIMENT SETUP.

Our method is implemented with python 3.8 and PyTorch 1.11. We conduct all experiments on a Ubuntu 20.04 server equipped with 64 CPUs and six Quadro RTX 6000 GPUs.

**Datasets and models.**    Two publicly available datasets (i.e., CIFAR-10 (Krizhevsky et al., 2009) and ImageNet (Russakovsky et al., 2015)) are used to evaluate the effectiveness of UNICORN. The details of the datasets can be found in the §A.4. Nine different network architectures (i.e., NiN (Lin et al., 2014), VGG16 (Simonyan & Zisserman, 2015), ResNet18 (He et al., 2016), Wide-ResNet34 (Zagoruyko & Komodakis, 2016), MobileNetV2 (Sandler et al., 2018), InceptionV3 (Szegedy et al., 2016), EfficientB0 (Tan & Le, 2019), DenseNet121, and DenseNet169 (Huang et al., 2017)) are used in the experiments. These architectures are representative and are widely used in prior backdoor related researches (Liu et al., 2019; Wang et al., 2019; Li et al., 2021d;c; Liu et al., 2022b; Xu et al., 2021; Huang et al., 2022; Xiang et al., 2022).

**Attacks.**    We evaluate UNICORN on the backdoors injected in different spaces, including pixel space, signal space, feature space and numerical Space. For pixel space, we use three different attacks (i.e., Patch (Gu et al., 2017), Blended (Chen et al., 2017), and SIG (Barni et al., 2019)). For signal space, we use Filter attack (Liu et al., 2019) that modifying the whole image with three different filters from Instagram, i.e., 1977, kelvin, and moon. For feature space, warping-based attack WaNet (Nguyen & Tran, 2021) is used. We also evaluate UNICORN on numbel space attack BppAttack (Wang et al., 2022c). More details about the used attacks can be found in the §A.5.

**Baselines.** We use four existing trigger inversion methods as baselines, i.e., Neural Cleanse (Wang et al., 2019), K-arm (Shen et al., 2021), TABOR (Guo et al., 2020) and Topological (Hu et al., 2022). Note that most existing trigger inversion methods use the optimization problem proposed in Neural Cleanse and share the same limitations (i.e., they can not generalize to different types of triggers).

**Evaluation metrics.** We use Attack Success Rate of the inverted trigger (ASR-Inv) on testset (the samples that are unseen during the trigger inversion) to evaluate the effectiveness of the trigger inversion methods. It is calculated as the number of samples pasted with inverted trigger that successfully attack the models (i.e., control the model to predict backdoor samples as the target label) divided by the number of all samples pasted with inverted trigger. High ASR is a fundamental goal of backdoor

Table 1: ASR-Inv on different attacks, models and datasets.

| Dataset | Network | Pixel Space | | | Signal Space | | | Feature | Numerical |
|---|---|---|---|---|---|---|---|---|---|
| | | Patch | Blend | SIG | 1977 Filter | Kelvin Filter | Moon Filter | WaNet | BppAttack |
| CIFAR-10 | NiN | 97.62% | 91.41% | 98.39% | 97.69% | 97.96% | 96.87% | 92.10% | 96.68% |
| | VGG16 | 99.86% | 98.83% | 98.43% | 97.42% | 99.60% | 98.72% | 98.71% | 91.97% |
| | ResNet18 | 99.61% | 99.74% | 99.17% | 96.54% | 94.57% | 97.50% | 99.95% | 99.33% |
| | Wide-ResNet34 | 98.51% | 98.52% | 94.14% | 95.31% | 99.64% | 97.73% | 84.45% | 91.91% |
| | MobileNetV2 | 96.50% | 99.48% | 95.31% | 93.51% | 96.60% | 93.12% | 98.75% | 82.29% |
| | InceptionV3 | 99.24% | 99.35% | 97.65% | 98.46% | 93.59% | 97.65% | 96.98% | 92.26% |
| | EfficientB0 | 98.95% | 97.89% | 95.99% | 92.42% | 95.54% | 91.41% | 85.93% | 95.89% |
| | DenseNet121 | 99.01% | 90.63% | 99.60% | 98.04% | 94.14% | 96.79% | 100.00% | 96.74% |
| ImageNet | VGG16 | 95.12% | 91.75% | 95.75% | 99.02% | 98.85% | 99.34% | 99.85% | 96.18% |
| | ResNet18 | 98.25% | 97.12% | 93.00% | 97.25% | 98.65% | 94.37% | 99.74% | 99.50% |
| | DenseNet169 | 92.45% | 87.75% | 85.50% | 93.85% | 97.93% | 96.28% | 94.62% | 90.45% |

attacks. The Attack Success Rate of the injected trigger is denoted as ASR-Inj. In appendix, we also show the results for different metrics, i.e., the backdoor models detection accuracy (§A.7) and the cosine similarity (SIM) between model's intermediate representations produced by injected triggers and the inverted triggers (§A.8).

## 5.2 EFFECTIVENESS.

To measure the effectiveness of UNICORN, we collect the ASR of the inverted triggers (ASR-Inv) generated by UNICORN and baselines on different attacks, datasets and models. Note that the ASR on the injected trigger (ASR-Inj) are all above 95%. We also show the visualizations of the inverted triggers. We assume only 10 clean samples for each class are available for trigger inversion as our default setting, which is a challenging but practical scenario. It is also a common practice. (Wang et al., 2019; Shen et al., 2021; Liu et al., 2019).

**Attack success rate of inverted triggers.** Table 1 shows the ASR-Inv under the backdoored models generated by attacks in different spaces. The inverted triggers generated by UNICORN achieves ASR-Inv in all settings. In detail, the average ASR-Inv under pixel space attacks, signal space attacks, feature space attacks and numerical space attacks are 96.37%, 96.55%, 95.55%, and 93.92%, respectively. These results indicate UNICORN is effective for inverting backdoor triggers injected in different spaces. We also show the visualization of the inverted triggers as well as the ground-truth triggers in Fig. 3. The first row shows the original image and the samples pasted with different injected triggers. The dataset and the model used here are ImageNet and VGG16. As can be observed, UNICORN can effectively invert the trigger that is similar to injected triggers. For example, for the backdoored models produced via patch attack (Gu et al., 2017), the inverted trigger share the same location (i.e., right upper corner) and the color (i.e., yellow) with the injected trigger. For the attack in filter space, feature space and numerical space, our method also successfully inverted the global perturbations. The inverted triggers might be not equal to the injected triggers. This is because existing backdoor attacks are inaccurate (Wang et al., 2022a). In other words, a trigger that is similar but not equal to the injected one is also able to activate the malicious backdoor behaviors of the model. We observed that pixel space attacks are more accurate than the attacks in other spaces. The results show that UNICORN can effectively invert the triggers in different input spaces.

**Comparison to existing methods.** To compare the effectiveness of UNICORN and existing trigger inversion methods, we collect the ASR-Inv obtained by different methods.

Table 2: ASR-Inv for different methods.

The dataset and the model used is CIFAR-10 and ResNet18, respectively. Results in Table 2 demonstrate the average ASR-Inv of UNICORN are 99.51%, 96.20%, 99.95%, and 99.33% for the attack in pixel space, signal space, feature space and numerical space, respectively. On average, the ASR-Inv of UNICORN is 1.32, 1.33, 1.34, and 1.30 times

| Space | Attack | NC | K-arm | TABOR | TOPO | UNICORN |
|---|---|---|---|---|---|---|
| Pixel | Patch | 92.40% | 89.47% | 94.57% | 92.48% | 99.61% |
| | Blend | 90.75% | 88.24% | 90.32% | 91.74% | 99.74% |
| | SIG | 89.69% | 91.32% | 86.93% | 89.33% | 99.17% |
| Signal | 1977 | 63.63% | 65.28% | 68.69% | 69.94% | 96.54% |
| | Kelvin | 67.46% | 63.54% | 65.02% | 67.24% | 94.57% |
| | Moon | 73.93% | 72.93% | 68.01% | 70.87% | 97.50% |
| Feature | WaNet | 61.90% | 62.50% | 64.86% | 62.06% | 99.95% |
| Numerical | BppAttack | 55.83% | 59.84% | 49.01% | 60.18% | 99.33% |

higher than that of Neural Cleanse (NC), K-arm, TABOR and Topological (TOPO), respectively. For existing methods, their inverted trigger has higher ASR-Inv on pixel space attacks, while the

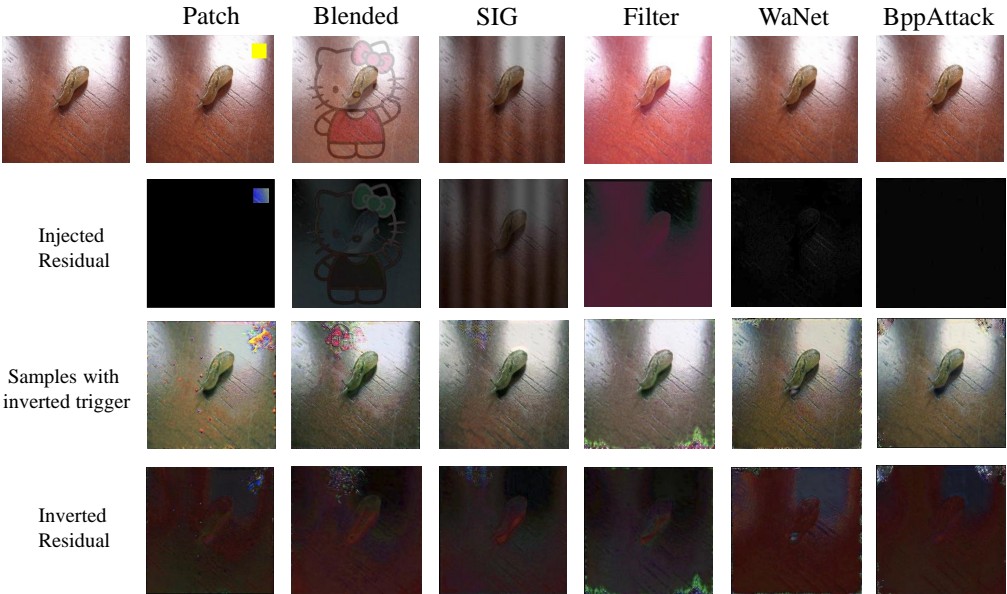

Fig. 3: Visualizations of the inverted triggers.

ASR-Inv on other three spaces is much lower. This is understandable because existing methods assumes the trigger is a static pattern with small size in the pixel space, and it formalize the trigger by using pixel space masks and patterns. However, UNICORN formalizes the triggers in a more general form, thus it has better generalizability and it is highly effective for various types of triggers.

## 5.3 ABLATION STUDIES.

In this section, we evaluate the impacts of different constraints used in Eq. 4 (i.e., stealthiness constraint and disentanglement constraint). More ablation studies can be found in §A.8.

**Effects of disentanglement constraint.** In our trigger inversion framework, we also add a constraint to disentangle the compromised activation vector and benign activation vector (§4). To investigate the effects of it, we compare the ASR-Inv of our method with and without the disentanglement constraint. The model and the dataset used is EfficientNetB0 and CIFAR-10. Results in Table 3 show that the ASR-Inv of the inverted triggers drop rapidly if the disentanglement constraint is removed. In detail, the average ASR-Inv is 94.25% and 64.56% for the method with and without disentanglement constraint, respectively. The results demonstrate that the disentanglement constraint is important for inverting the injected triggers in the model.

**Effects of stealthiness constraint.** As we discussed in §4, we use SSIM score to constrain the stealthiness of the backdoor samples. In Eq. 4, we use a threshold value $\gamma$ to constrain the SSIM score. The higher the SSIM score is, the stealthier the trigger is. In this section, we show the inverted backdoor samples generated by the framework with and without the SSIM constraint (stealthiness constraints). The results are demonstrated in Fig. 5e. We also show the sample pasted with injected trigger at the left

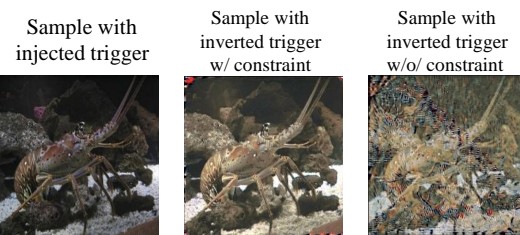

Fig. 4: Effects of stealthiness constraint.

most column. As can be observed, when we remove the stealthiness constraint, the main contents in the image becomes difficult to tell, and the quality of the image becomes low. When the constraint is added, the samples pasted with the inverted trigger is close to the injected backdoor sample. Thus, to satisfy the stealthiness requirement, adding the stealthiness constraint in the optimization of trigger inversion is needed. We set $\gamma = 0.85$ as the default setting.

## 5.4 EXTENSION TO SELF-SUPERVISED MODELS.

To investigate if UNICORN can generalize to models generated by different training paradigms, in this section, we evaluate our framework on self-supervised learning models. Different from super-

Table 3: Effects of disentanglement constraint.

| | Pixel | | | Signal | | | Feature | Numerical |
|---|---|---|---|---|---|---|---|---|
| | Patch | Blended | SIG | 1977 Filter | Kelvin Filter | Moon Filter | WaNet | BppAttack |
| ASR-Inj | 99.92% | 99.70% | 98.71% | 98.46% | 99.31% | 98.67% | 98.69% | 99.18% |
| ASR-Inv w/o Disentanglement | 68.08% | 64.64% | 68.78% | 63.20% | 55.42% | 70.50% | 58.82% | 70.54% |
| ASR-Inv w/ Disentanglement | 98.95% | 97.89% | 95.99% | 92.42% | 95.54% | 91.41% | 85.93% | 95.89% |

vised learning, self-supervised learning (Chen et al., 2020; He et al., 2020) first pre-trains an encoder with unlabeled data, and then builds downstream classifiers by using the encoder as feature extractor. In backdoor attacks for self-supervised learning, backdoor injected into pre-trained encoders can be inherited to the downstream classifiers. Directly conducting trigger inversion on encoders is important as it can reduce the cost of training extra downstream classifiers for inverting triggers. In this section, we use a mainstream self-supervised learning backdoor attack Badencoder (Jia et al., 2022) to evaluate if UNICORN can be applied for self-supervised models. Badencoder assumes that attackers have knowledge of the downstream task and have access to few reference samples belong to the downstream target class. It first trains a clean encoder using SimCLR (Chen et al., 2020), then injects backdoors into this pre-trained encoder. By default, Badencoder uses a white square patch as its trigger. Its backdoor optimization maximizes the similarity between clean encoder extracted and backdoored encoder extracted features of benign inputs, and maximizes the similarity between backdoored encoder extracted features of inputs with triggers and features of reference samples.

We use the backdoored encoder generated by Badencoder on CIFAR-10 dataset (Krizhevsky et al., 2009) to invert triggers. Instead of minimizing the classification loss on the target labels, we use one reference sample for each target class as the Badencoder does, and maximize the similarity between the embedding of reference sample and that of samples pasted with inverted triggers. As illustrated in Fig. 6, we visualize the inverted triggers and injected triggers in a backdoored encoder. It shows

Table 4: ASR in self-supervised learning.

| Target class | ASR-Inj | ASR-Inv |
|---|---|---|
| Airplane | 98.92% | 99.92% |
| Truck | 99.81% | 99.75% |
| Horse | 94.88% | 91.35% |
| Average | 97.87% | 97.01% |

that UNICORN can invert a similar patch trigger as the injected one at the bottom right corner of input images. We also evaluate the attack success rate of our inverted triggers (ASR-Inv) in downstream classifiers. We adopt STL10 (Coates et al., 2011) as the downstream dataset and fine-tune 20 epochs for classifiers. The target classes of the reference input we use are truck, airplane, and horse, respectively, which are the same as Badencoder. In Table 4, ASR-Inj means the ASR of injected triggers. From Table 4, we can observe that in different target class, our inverted triggers all have comparative ASR (i.e., over 90%) as injected triggers. Similar to Neural Cleanse (Wang et al., 2019), the ASR-Inv is even larger than ASR-Inj in some cases. It is not surprising given the trigger is inverted by using a scheme that optimizes for higher ASR.

## 6 DISCUSSION

**Ethics.** Studies on adversarial machine learning potentially have ethical concerns. This paper proposes a general framework to invert backdoor triggers in DNN models. We believe this framework can help improve the security of DNNs and be beneficial to society.

**Extention to NLP models.** In this paper, we have discussed backdoor triggers in different input spaces in CV models. Similarly, backdoor triggers can be injected into NLP models in different input spaces, e.g., word space (Chen et al., 2021; Qi et al., 2021b), token space (Shen et al., 2021; Zhang et al., 2021), and embedding space (Qi et al., 2021a; Kurita et al., 2020; Yang et al., 2021). Expanding our framework to NLP trigger inversion will be our future work.

## 7 CONCLUSION

In this paper, we formally define the trigger in backdoor attacks against DNNs, and identify the inner behaviors in backdoor models based on the analysis. Based on it, we propose a backdoor trigger inversion method that can generalize to various types of triggers. Experiments on different datasets, models and attacks show that our method is robust to different types of backdoor attacks, and it outperforms prior methods that are based on attack-specific constraints.

ACKNOWLEDGEMENT

We thank the anonymous reviewers for their valuable comments. This research is supported by IARPA TrojAI W911NF-19-S-0012. Any opinions, findings, and conclusions expressed in this paper are those of the authors only and do not necessarily reflect the views of any funding agencies.

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

# A APPENDIX

## A.1 DEFINITIONS FOR §3.3.

In this section, we provide definitions for the notations used in §3.3.

**Definition 2** *(Sub-models) Given a neural network $\mathcal{M}$, we have $h$ and $g$ are the sub-models from the input layer to the intermediate layer and the sub-model from the intermediate layer to the output layer, respectively.*

Intuitively, a neural network can be split into two functions. There are different ways to split the model. In this paper, we split the model at the last convolutional layer.

**Definition 3** *(Activation Vector) Given a neural network $\mathcal{M} = g \circ h$, where $h$ and $g$ are the sub-models, we define the activation vector $A$ as the output of function $h$, i.e., $A = h(x)$, where $x$ is the input of the model.*

**Definition 4** *(Unit Vector) Given a neural network $\mathcal{M} = g \circ h$, where $h$ and $g$ are the sub-models, we define $u$ as the unit vector (i.e., $|u| = 1$) in the intermediate representation space $N$ mapped by function $h$, where $I \xrightarrow{h} N$, $I$ is the input space of the model. The dimensions of $u$ are identical to $h(x)$, where $x$ is the input of the model.*

**Definition 5** *(Projection Length) Given a neural network $\mathcal{M} = g \circ h$, where $h$ and $g$ are the sub-models, we define $u(x)$ as the projection length on unit vector $u$ for the intermediate representation of input $x$, i.e., $u(x) = \frac{h(x) \cdot u}{|u|}$. The projection length $u(x)$ is a scalar.*

## A.2 SUPPORT FOR COROLLARY 1.

In this section, we provide more support for Corollary 1, which describes the invariant of the backdoor attacks. It can be supported by the findings in existing researches, as well as our empirical evidences. Liu et al. (2019) show that some specific neurons (compromised neurons) can represent the trigger feature that can significantly elevate the probability of the target label when their activation value is set to a narrow region, which can support our observation. We also conducted experiments on CIFAR-10 (Krizhevsky et al., 2009) and ResNet18 (He et al., 2016) with Patch attack (Gu et al., 2017) to verify it. We first calculated the compromised activation vector $A_c$ and benign activation vector $A_b$ based on their definitions. We then perturb $A_b$ by using 100 randomly sampled clean inputs. After that, we feed the perturbed $A_b$ and calculate $A_c$ into the submodel $g$, and compute the attack success rate, i,e., $P(g(A_c, A_b) = y_t)$. The results show that the attack success rate is 95.32% even though the benign activation values $A_b$ is perturbed. The results demonstrate perturbing $A_b$ will not influence the backdoor behaviors of the infected model, namely, $A_c$ and $A_b$ are disentangled.

## A.3 DISCUSSION ABOUT EXISTING WORKS ABOUT BACKDOOR BEHAVIORS

Existing works also find some backdoor related behaviors, such as having a shortcut path when inferring (Zheng et al., 2021; Li et al., 2022). However, they can not be integrated with our optimization. For Zheng et al. (2021), such "shortcut" behavior is modeled by the neuron connectivity graph and its topological features of the model. The topology features are model-specific, and the time cost for extracting them is high. In detail, the runtime of extracting the neuron connectivity graph and its topological features for each ResNet18 (He et al., 2016) model on the CIFAR-10 (Krizhevsky et al., 2009) dataset is 373s while the time cost for computing our disentanglement loss is only 0.36s. Integrating the "shortcut" constraint in our optimization requires conducting the topological feature extraction for each optimization step, which will take a significantly longer time.

## A.4 DETAILS OF DATASETS

In this section, we introduce the details of the datasets involved in the experiments. All datasets used are open-sourced.

**CIFAR10 (Krizhevsky et al., 2009).** This dataset is built for recognizing general objects such as dogs, cars, and planes. It contains 50000 training samples and 10000 training samples in 10 classes.

**ImageNet (Russakovsky et al., 2015).** This dataset is a large-scale object classification benchmark. In this paper, we use a subset of the original ImageNet dataset specified in Li et al. (Li et al., 2021d). The subset has 100000 training samples and 10000 test samples in 200 classes.

## A.5 DETAILS OF ATTACKS

In this section, we introduce the details of the used backdoor attacks. The attacks are in single-target setting. The target label of the attack is randomly selected for each backdoored model.

**Patch Attack (Gu et al., 2017).** This attack uses a static pattern (i.e., a patch) as backdoor trigger. It generates backdoor inputs by pasting its pre-defined trigger pattern (e.g., a colored square) on the original inputs. It then compromises the victim models by poisoning a subset of the training data (i.e., injecting backdoor samples and modifying their labels to target labels). In our experiments, we use a $3 \times 3$ yellow patch located at the right-upper corner as the backdoor trigger. The poisoning rate we used is 5%.

**Blend Attack (Chen et al., 2017).** This attack generates backdoor samples via blending the clean input with a predefined pattern, e.g., a cartoon cat. It also considers the poisoning threat model where the attacker can modify the training data, but can not control the whole training process. The poisoning rate we used is 5%.

**SIG Attack (Barni et al., 2019).** This attack uses superimposed sinusoidal signals as backdoor triggers. It assumes that adversary can poison a subset of training samples but can not fully control the training process. In our experiments, we set the poisoning rate as 5%. Also, we set the frequency and the magnitude of the backdoor signal as 6 and 20, respectively.

**Filter Attack (Liu et al., 2019).** This attack exploits image filters as triggers and creates backdoor samples by applying selected filters on images. Similar to BadNets, the backdoor triggers are injected with poisoning. In our experiments, we use a 5% poisoning rate and apply the 1977, Kelvin and Moon filter from Instagram as backdoor triggers.

**WaNet (Nguyen & Tran, 2021).** This method achieves backdoor attacks by image warping. It adopts an elastic warping operation to transform backdoor triggers. Different from BadNets and Filter Attack, in WaNet, the adversary can modify the training process of the victim models to make the attack more resistant to backdoor defenses. WaNet is nearly imperceptible to human, and it is able to bypass many existing backdoor defenses (Gao et al., 2019; Chen et al., 2018; Liu et al., 2018a; Wang et al., 2019). In our experiments, the warping strength and the grid size are set to 0.5 and 4, respectively.

**BppAttack (Wang et al., 2022c).** This attack creates backdoor samples by using image quantization and dithering techniques. It changes the numerical representations of the images. It considers the attacker that can access the full control of the training process. It is even more stealthier than WaNet (Nguyen & Tran, 2021). In this paper, we set the bits depth for the backdoor inputs as 5.

## A.6 COEFFICIENT ADJUSTMENT IN OPTIMIZATION

In this section, we discuss the detailed method for adjusting the coefficients for different items in the optimization. Following existing works (Wang et al., 2019; Liu et al., 2022b), coefficients are dynamic in the optimization. When the loss value is larger (or smaller) than the threshold value (i.e., the loss value does not satisfy the constraint), we use a large coefficient $w_{large}$. Otherwise, if the loss value satisfy the constraint, we apply a small coefficient $w_{small}$. By default we set $w_{small} = 0$. The default $w_{large}$ value for $w_1$, $w_2$, $w_3$, and $w_4$ are 200, 10, 10, and 1.

## A.7 BACKDOOR MODELS DETECTION RESULTS

To compare the performance of UNICORN and existing methods on backdoor models detection task, we report the detailed true positive, false positive, false negative, true negative, and detection accuracy for different trigger inversion methods. In our method, we determine a model is infected

Table 5: Backdoor models detection results.

| Method | Pixel Space Attack | | | | | Signal Space Attack | | | | | Feature Space Attack | | | | |
|---|---|---|---|---|---|---|---|---|---|---|---|---|---|---|---|
| | TP | FP | FN | TN | Acc | TP | FP | FN | TN | Acc | TP | FP | FN | TN | Acc |
| NC | 19 | 1 | 1 | 19 | 95.0% | 2 | 1 | 18 | 19 | 52.5% | 8 | 1 | 12 | 19 | 67.5% |
| K-arm | 20 | 0 | 0 | 20 | 100.0% | 0 | 0 | 20 | 20 | 50.0% | 9 | 0 | 11 | 20 | 72.5% |
| TABOR | 20 | 3 | 0 | 17 | 92.5% | 5 | 3 | 15 | 17 | 55.0% | 3 | 3 | 17 | 17 | 50.0% |
| Hu et al. | 18 | 1 | 2 | 19 | 92.5% | 4 | 1 | 16 | 19 | 57.5% | 9 | 1 | 11 | 19 | 70.0% |
| UNICORN | 19 | 1 | 1 | 19 | 95.0% | 20 | 1 | 0 | 19 | 97.5% | 18 | 1 | 2 | 19 | 92.5% |

Table 6: Influence of hyperparameters.

| Attack | $\alpha$ | | | | $\beta$ | | | | | $\gamma$ | | | | | $\delta$ | | | | $p$ | | | |
|---|---|---|---|---|---|---|---|---|---|---|---|---|---|---|---|---|---|---|---|---|---|---|
| | 0.005 | 0.01 | 0.05 | 0.1 | 1% | 3% | 5% | 10% | 20% | 0.65 | 0.75 | 0.85 | 0.90 | 0.95 | 0.5 | 1 | 5.0 | 10.0 | 0.04% | 0.10% | 0.20% | 1.00% |
| Patch | 0.87 | 0.88 | 0.82 | 0.80 | 0.46 | 0.65 | 0.83 | 0.88 | 0.80 | 0.77 | 0.83 | 0.88 | 0.87 | 0.79 | 0.88 | 0.87 | 0.62 | 0.56 | 0.70 | 0.88 | 0.88 | 0.89 |
| 1977 | 0.72 | 0.72 | 0.70 | 0.68 | 0.38 | 0.60 | 0.69 | 0.72 | 0.59 | 0.66 | 0.69 | 0.72 | 0.73 | 0.60 | 0.72 | 0.73 | 0.50 | 0.47 | 0.61 | 0.66 | 0.72 | 0.72 |
| WaNet | 0.81 | 0.80 | 0.77 | 0.74 | 0.54 | 0.62 | 0.76 | 0.80 | 0.65 | 0.67 | 0.72 | 0.80 | 0.80 | 0.70 | 0.80 | 0.81 | 0.64 | 0.55 | 0.74 | 0.78 | 0.80 | 0.81 |
| BppAttack | 0.67 | 0.69 | 0.66 | 0.62 | 0.42 | 0.56 | 0.65 | 0.69 | 0.58 | 0.62 | 0.65 | 0.69 | 0.67 | 0.61 | 0.69 | 0.67 | 0.49 | 0.43 | 0.56 | 0.60 | 0.69 | 0.68 |

with a backdoor if the ASR-Inv of a label is larger than 90%. The dataset and the model used are CIFAR-10 (Krizhevsky et al., 2009) and ResNet18 (He et al., 2016). We conduct the experiments on attacks in different spaces, i.e., Patch (Gu et al., 2017), 1977 Filter (Liu et al., 2019), and WaNet (Nguyen & Tran, 2021). For each attack, we train 20 benign models and 20 backdoor models. The results demonstrate that our method achieves high detection accuracy under the attacks in different spaces. For the attacks in all spaces, the detection accuracy is higher than 90%. However, existing methods can only handle the attack in pixel space, and they have low detection accuracy for the attacks in other spaces.

## A.8 MORE ABLATION STUDIES

In this section, we study the influence of different hyperparameters, i.e., the threshold values $\alpha$, $\beta$, $\gamma$, and $\delta$ used in Eq. 4, as well as the portion of the required clean training samples $p$. The dataset and the model used is CIFAR-10 (Krizhevsky et al., 2009) and EfficientNetB0 (Tan & Le, 2019), respectively. We show the cosine similarity (SIM) between model's intermediate representations produced by injected triggers (ground-truth) and the inverted triggers under different hyper-parameters. A higher SIM value means the intermediate representations of the inverted trigger are closer to that of the injected trigger. As can be observed, the similarity between the inverted trigger features and the injected trigger features is stable when $\alpha < 0.01$, $0.85 < \gamma < 0.95$, and $\delta < 1.0$. These results show the stability of UNICORN. For $\beta$, in most cases, the SIM achieves the highest value when the value of is around 10% of the entire input space. For the portion of the required clean training samples $p$, the SIM value is low when the portion is smaller than 0.10% (5 samples per class). When it is larger than 0.20% (10 samples per class), the results are stable, and UNICORN has high SIM. In Table 7, we also show the SIM value for the patch triggers with different sizes under different $\beta$ value. For larger patch trigger, the SIM value is still high when is $\beta$ around 10% of the entire input space, showing the robustness of our method.

## A.9 EFFICIENCY

In this section, we discuss the efficiency of our method. We compared the runtime of our method and that of Neural Cleanse (Wang et al., 2019) on CIFAR-10 (Krizhevsky et al., 2009) and ResNet18 (He et al., 2016). The result shows that our runtime is 2.7 times of Neural Cleanse's runtime. We admit

Table 7: Influence of $\beta$ on patch trigger with different sizes.

| Patch Size | $\beta = 1\%$ | $\beta = 3\%$ | $\beta = 5\%$ | $\beta = 10\%$ | $\beta = 20\%$ |
|---|---|---|---|---|---|
| 3*3 | 0.46 | 0.65 | 0.83 | 0.88 | 0.80 |
| 6*6 | 0.46 | 0.62 | 0.78 | 0.82 | 0.81 |
| 9*9 | 0.44 | 0.55 | 0.69 | 0.77 | 0.79 |

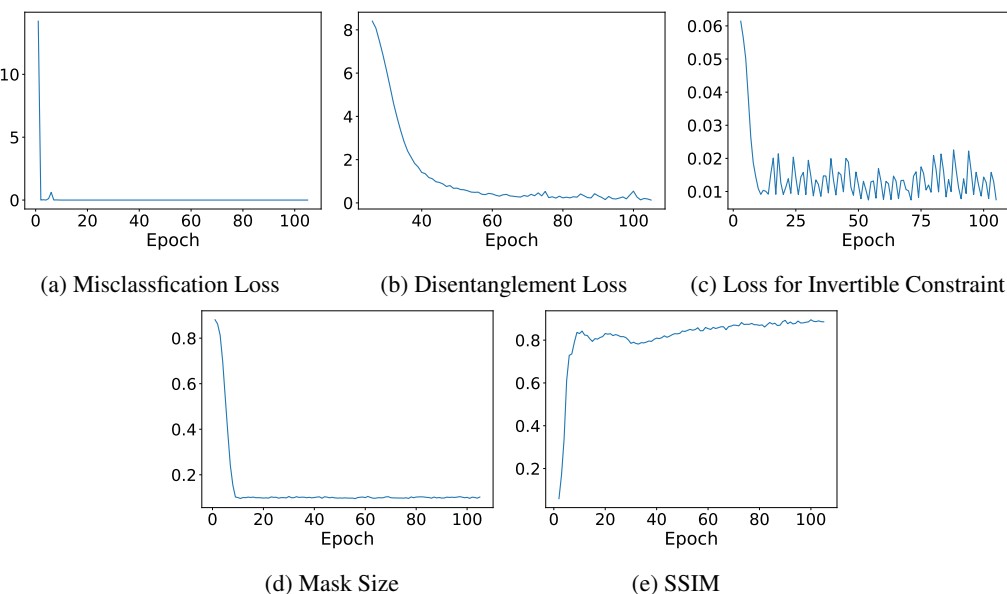

Fig. 5: Loss values during the optimization process.

Table 8: Results of integrating disentanglement constraint with NC.

| Method | ASR-Inv | Detection Accuracy |
|---|---|---|
| NC | 63.63% | 52.5% |
| NC+Distenglement | 70.35% | 55.0% |

the computational complexity of our method is larger than existing methods. However, our method is more general and robust for inverting different types of backdoor triggers, while existing methods can only handle pixel space triggers. In addition, our method can be accelerated by existing works K-arm scheduler (Shen et al., 2021) and mixed precision training (Micikevicius et al., 2017).

### A.10 STABILITY OF OUR OPTIMIZATION

In this section, we discuss the stability of our optimization. Since our method optimize several components simultaneously, the discussion about the stability of our optimization process is meaningful. Our optimization is stable, and it can be supported by existing works. The optimization processes for existing backdoor attacks are also complicated. Many existing backdoor attacks (e.g., Cheng et al. (2021), Doan et al. (2021b), Doan et al. (2021a)) use generative models as backdoor triggers. In these attacks, the backdoors are injected by simultaneously optimizing the generative networks and the victim models. For example, Cheng et al. (2021) optimize the victim model and a complex CycleGAN (Zhu et al., 2017) at the same time to inject the backdoor. Such processes also have many parameters to optimize. Their training loss can reduce smoothly, and they can get high ASR and benign accuracy. These results give us more confidence to invert backdoor triggers via optimizing generative models. In Fig. 5, we show the plot of training loss values for different items in Eq. 5, i.e., misclassification objective $\mathcal{L}\left(\mathcal{M}(\tilde{\boldsymbol{x}}), y_t\right)$ (Fig. 5a), disentanglement loss $\mathcal{L}_{dis}$ (Fig. 5b), loss for invertible constraint $\|Q(P(\boldsymbol{x})) - \boldsymbol{x}\|$ (Fig. 5c), mask size $\|\boldsymbol{m}\|$ (Fig. 5d), and SSIM between the benign samples and inverted samples SSIM$(\tilde{\boldsymbol{x}}, \boldsymbol{x})$ (Fig. 5e). The X-axis is the epoch number, and the Y-axis denotes the loss value. The dataset, model and attack used are CIFAR-10 (Krizhevsky et al., 2009) and ResNet18 (He et al., 2016), and 1977 Filter (Liu et al., 2019). Results demonstrate our the loss values are smoothly reduced during the optimization, showing the stability of our method.

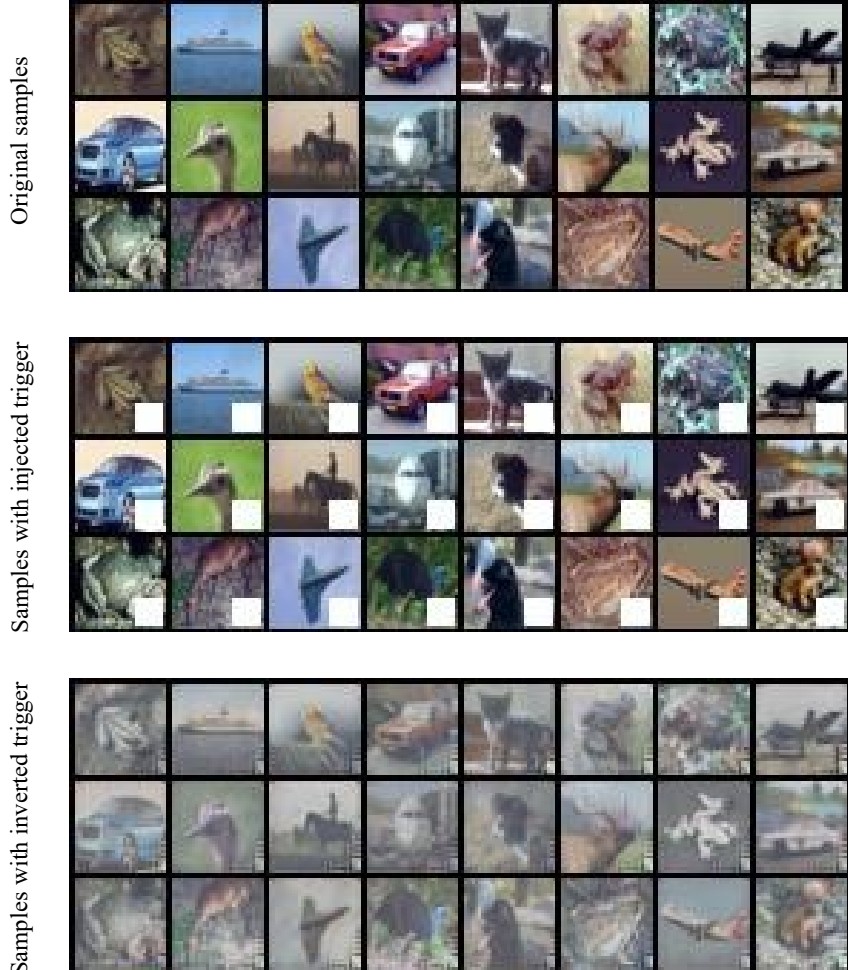

Fig. 6: Visualization of inverted triggers of backdoored encoders.

## A.11 RESULTS FOR INTEGRATING DISENTANGLEMENT CONSTRAINT WITH EXISITING METHODS.

To study the effects of the disentanglement constraint when integrate it with existing methods, we conduct the experiment of adding the disentanglement constraint on Neural Cleanse (NC) (Wang et al., 2019). The attack, dataset, and model used is 1977 Filter (Liu et al., 2019), CIFAR-10 (Krizhevsky et al., 2009), and ResNet18 (He et al., 2016), respectively. The results are shown in Table 8. It demonstrates that the ASR-Inv and the detection accuracy are still low when adding the disentanglement constraint on NC. This is because the limitation of the existing formulation in the inversion problem is that it can not handle the trigger not injected in the pixel space. Adding disentanglement constraint on it can not solve the limitation. However, the invertible transformation given in Eq. 4 makes inverting the trigger in different spaces possible.

