# OpenReview forum: "UNICORN: A Unified Backdoor Trigger Inversion Framework"
_ICLR.cc/2023/Conference — ICLR 2023 notable top 25%_

### Official Review · Reviewer_8Hde · 2022-10-17

**Confidence:** 4
**Correctness:** 3
**Technical Novelty And Significance:** 3
**Empirical Novelty And Significance:** 3
**Recommendation:** 6

**Clarity, Quality, Novelty And Reproducibility:**

Detailed comments are given in the **Strength And Weaknesses**. Below is a summary:

#### **Clarity**: as mentioned in the weaknesses above, several parts, such as the proposed method, need to be rewritten to help with the clarity.

#### **Quality**: overall, the paper is of moderate quality. Although the work has some positive aspects (such as the motivation for looking into the inversion problem in a different space than the pixel space), the motivations behind different parts of the methodology are obscure. Moreover, the experimental results require a deeper investigation and a broader set of baselines.

#### **Novelty And Reproducibility**: while the proposed method seems novel, certain aspects of the work need proper motivation. In terms of reproducibility, the authors provide the codebase. However, a detailed account of the hyper-parameter selection is missing. For instance, while $\gamma$ and $\delta$ values were given on page 5, it is unclear how those specific values were found.

**Strength And Weaknesses:**

### Strengths:
- The general idea of the paper is novel and interesting. Specifically, since the previous methods would only use a fixed objective function in the pixel space, they lack the universality to encompass all the existing backdoor triggers. UNICORN seems to be a step in the right direction.

 - A broad range of backdoor attacks and neural network architectures are used in the experimental results.


### Weaknesses:

The weaknesses can be divided into three broad categories:
#### **1. Motivation and Methodology**:
- The motivation behind the proposed approach during the introduction seems clear. However, as the reader goes through Section 3, some aspects start to be introduced without proper motivation. For instance, while the material presented in Section 3.3 makes sense, a reader might wonder why this information is presented there.

- The issue with the material presented in Section 3.3 shows itself more profoundly in Eqs. (1) and (2). A reader might say: "Okay, based on the material, I can understand that we need to map the image to a different space, but how did we get to a point where we need the *disentanglement constraint*? What was the authors' motivation behind adding this regularizer? More broadly, among all the regularizers in the world that one could have picked about backdoor attacks, why this particular one? Because similar types of properties about backdoor attacks have already been observed in the literature (e.g., backdoored models have a shortcut path when inferring backdoored examples [1, 2]). What is so specific about the proposed *disentanglement constraint* that makes it vital to the success of UNICORN?"

- In the leadup to Section 3, the paper mentions several times that "...[existing methods] assume that the trigger is a static pattern with a *small size* in the pixel space." Nevertheless, for Eq. (1) in Section 4, the paper says that: "similar to Neural Cleanse, we also constrain the size of the input space mask." These seem to contradict each other, as the paper argues that the assumption of having a small-size trigger is problematic, yet the same constraint is used!

- The motivation behind trigger inversion needs to be more solid. The reader needs to understand what is the advantage of reverse-engineering the trigger. Otherwise, they could argue that a simple detection-based strategy can tell us if the model has backdoors or not, and we might not need the hassle of finding the underlying trigger.

- In the abstract, introduction, and background, it is mentioned that: "...existing work does not formally define the trigger and the inversion problem," which can create misinterpretations as those methods define the trigger inversion problem in their own mathematical ways. It would be better to replace these with something like: "...existing work does not consider the trigger's design space in their formulation of the inversion problem."

#### **2. Presentation and Writing**:
- Besides the motivational issues about the origins of the *disentanglement constraint* mentioned above, presentation of the paper, especially Section 4, needs lots of work. The reader suddenly faces an objective function in Eq. (1), where most of the terms have not been introduced already, and their definitions are deferred to the bottom of the page. Perhaps reasoning about different properties of the sought trigger and then presenting the mathematical expressions would be a better idea and makes the paper a lot easier to read and follow.

#### **3. Experimental Results**:
- The number of baselines used in the paper is limited. More recent trigger inversion baselines, such as [3], need to be added.

- While the experimental results show that the reverse-engineered triggers can have a high attack success rate, they do not tell us the other side of the story: false positives. It is common practice to show whether the trigger inversion helps detect backdoor models and report the detection accuracy among models. In other words, given a set of clean and poisoned models, how many instances could the proposed method identify the trigger correctly? How many clean cases still were identified as malicious?

- Since the model uses a trainable neural network for trigger inversion, it would be beneficial to see how it compares to existing methods (such as Neural Cleanse) in terms of the detection time and computational complexity.

- Ablation studies are limited. First, it is unclear what benchmark and dataset were used for the "effects of the mask size constraint." Secondly, the results in Table 3 need to be repeated for different triggers, especially those with large sizes. Finally, given the importance of the *disentanglement constraint*, what would happen if one adds this regularizer to the existing methods such as Neural Cleanse? Can this be used to boost their performance? If so, what is the role of the invertible transformation given in Eq. (1)?

- In Table 5, how is the reverse-engineered trigger's success rate better than the original one?

- The effects of different hyper-parameters and how they are set seems missing.

[1] Zheng et al. "Topological detection of Trojaned neural networks," *NeurIPS*, 2021.

[2] Li et al. "Deep learning backdoors," *Security and Artificial Intelligence*, *Springer*, 2022.

[3] Hu et al. "Trigger hunting with a topological prior for trojan detection," *ICLR*, 2022.

**Summary Of The Paper:**

This paper proposes a backdoor inversion method called UNICORN. This approach is motivated by the fact that some backdoor attacks would apply their triggers in a space other than the pixel space. To alleviate this issue, UNICORN employs an objective function that uses an invertible transformation to map the image to a different space and apply the trigger. The goal is to generalize the previous trigger inversion methods that would only consider triggers applied in the pixel space. Experimental results indicate the success of the proposed approach in creating triggers that have similar characteristics to the original triggers and successfully fool the backdoored model.

**Summary Of The Review:**

Based on the grounds given above, the paper seems to require a better explanation of the method and better conduction of experiments. Thus, this reviewer would recommend rejection at this stage but would be happy to increase the rating if the authors can address the abovementioned concerns.

---

> ### Author Response · Authors · 2022-11-12
> **Response to Reviewer 8Hde - Part 1**
>
> Thank you very much for your time and insightful comments. We are encouraged the
> novelty of our work is recognized. We hope the following new clarifications and
> results can address your concerns. We are happy to answer more questions and conduct
> more experiments if needed.
>
> **Q1:** The motivation of the material presented in Section 3.3 is unclear. How
> did we get to a point where we need the disentanglement constraint? What was the
> authors' motivation behind adding this regularizer?
>
> **A1:** Thanks for your insightful comments. Directly optimizing without the
> disentanglement constraint can not guarantee that the inverted transformations have
> trigger effects (i.e., when the trigger features are extracted, the model will
> predict the target label regardless of the benign features). It will yield general
> transformations which are not specific to the injected backdoor. Thus, we need to
> add the constraint to ensure the transformation extracted by the neural network is
> meaningful and corresponds to the injected backdoor trigger. This is the
> motivation for the disentanglement constraint. We will add more discussion and make
> it more clear.
>
> **Q2:** Among all the regularizers in the world that one could have picked about
> backdoor attacks, why this particular one? Similar types of properties about
> backdoor attacks have already been observed in the literature (e.g., backdoored
> models have a shortcut path when inferring backdoored examples). What is so
> specific about the proposed disentanglement constraint that makes it vital to
> the success of UNICORN?
>
> **A2:** We found the injected trigger has some invariant in the intermediate
> representation of the victim model, and we formalize it as the disentanglement
> constraint. We found using the disentanglement constraint can make the inverted
> trigger closer to the injected one and have higher ASR-Inv. The reason why we
> use the proposed disentanglement constraint is that we believe it models the
> invariant of the backdoor trigger better. We agree that existing works also find
> some backdoor related behaviors, such as having a shortcut path when inferring
> (Zheng et al., Li et al.). However, they can not be integrated with our
> optimization. For Zheng et al., such "shortcut path" behavior is modeled by the
> neuron connectivity graph and its topological features of the model. The
> topology features are model-specific, and the time cost for extracting them is
> high. In detail, the runtime of extracting the neuron connectivity graph and its
> topological features for each ResNet18 model on the CIFAR-10 dataset is 373s
> while the time cost for computing our disentanglement loss is only 0.36s.
> Integrating the "shortcut" constraint in our optimization requires conducting
> the topological feature extraction for each optimization step, which will take a
> significantly longer time. We will add more discussion for existing works
> about the "shortcut path" behavior in the revised version.
>
> Zheng et al., Topological detection of Trojaned neural networks. NeurIPS, 2021.
>
> Li et al., Deep learning backdoors. Security and Artificial Intelligence, Springer, 2022
>
> **Q3:** In the leadup to Section 3, the paper mentions several times that
> "...[existing methods] assume that the trigger is a static pattern with a small
> size in the pixel space." Nevertheless, for Eq. (1) in Section 4, the paper says
> that: "similar to Neural Cleanse, we also constrain the size of the input space
> mask." These seem to contradict each other, as the paper argues that the
> assumption of having a small-size trigger is problematic, yet the same
> constraint is used!
>
> **A3:** Thanks for your helpful question. Note that there exist different input
> spaces, such as pixel space, signal space, and feature space. We want to clarify
> that our method does not constrain the same mask constraint of Neural Cleanse in
> the pixel space. Instead, we constrain the mask size in a specific input space
> introduced by transformation function $\phi$. We also want to clarify that
> we did not mean constraining the mask size in input space is problematic. Our
> point is the assumption of having a small-size trigger in the pixel space, and
> directly constraining the mask size in the pixel space is not valid for the
> triggers injected into other input spaces (e.g., signal space and feature
> space). Based on the Stealthiness Goal of the backdoor attack, the perturbations
> in the specific input space caused by the trigger can not be too large and the
> semantics of the benign content should be preserved. Therefore, we also need to
> constrain the mask size. We will revise it to make it clear.

---

> > ### Author Response · Authors · 2022-11-12
> > **Response to Reviewer 8Hde - Part 2**
> >
> > **Q4:** The motivation behind trigger inversion needs to be more solid.
> >
> > **A4:** Thanks for your constructive comment. The motivation behind trigger
> > inversion can be summarized as follows. We will add more discussion and revise
> > accordingly.
> >
> > * Besides determining whether a given model is backdoored or not, trigger inversion
> > methods can also provide more useful information and have many further
> > applications such as filtering out backdoor samples and mitigating the backdoor
> > injected in the models. For example, existing trigger inversion method Neural
> > Cleanse shows the defender can mitigate the backdoor by conducting "backdoor
> > unlearning" on the inverted trigger.
> >
> > * In addition, trigger inversion is a popular and well-established research
> > line. It achieves very good results in many existing research papers (e.g., Wang
> > et al., Liu et al., Guo et al., Shen et al., and Hu et al.) and competitions
> > (e.g., NIST TrojAI Competition), showing it is a promising direction.
> >
> > Wang et al., Neural Cleanse: Identifying and Mitigating
> > Backdoor Attacks in Neural Networks. IEEE S&P 2019.
> >
> > Liu et al., ABS: Scanning Neural Networks for Back-doors by
> > Artificial Brain Stimulation. CCS 2019.
> >
> > Guo et al., TABOR: A Highly Accurate Approach to Inspecting and
> > Restoring Trojan Backdoors in AI Systems. ICDM 2020.
> >
> > Shen et al., Backdoor Scanning for Deep Neural Networks through K-Arm Optimization. ICML 2021.
> >
> > Hu et al., Trigger Hunting with a Topological Prior for Trojan Detection. ICLR 2022.
> >
> > **Q5:** Definition of trigger in existing works.
> >
> > **A5:** Thanks for your insightful comment. We will revise it to "...existing
> > work does not consider the trigger's design space in their formulation of the
> > inversion problem.".
> >
> > **Q6:** Writing of Section 4 needs to be improved.
> >
> > **A6:** Thanks for your valuable comment and suggestions. We will reorgnized
> > Section 4 accordingly to make it easier to read and follow.
> >
> > **Q7:** The number of baselines used in the paper is limited.
> >
> > **A7:** Thanks for your insightful comment. The results of the comparison of
> > ASR-Inv with more baselines (i.e., TABOR and Hu et al.) can be found in the
> > following table. We will add the results to the revised version. We also report
> > the comparison of backdoored models detection accuracy as suggested. Please see
> > A8 for more details and results.
> >
> >  Attack| NC | K-arm | TABOR | Hu et al. | UNICORN
> >  ---|--- | ---| ---| ---| ---
> > Patch|92.40% | 89.47%| 94.57%| 92.48%| 99.61%
> > Blend|90.75% | 88.24%| 90.32%| 91.74%| 99.74%
> > SIG|89.69% | 91.32%| 86.93%| 89.33%| 99.17%
> > 1977|63.63% | 65.28%| 68.69%| 69.94%| 96.54%
> > Kelvin|67.46% | 63.54%| 65.02%| 67.24%| 94.57%
> > Moon|73.93% | 72.93%| 68.01%| 70.87%| 97.50%
> > WaNet|61.90% | 62.50%| 64.86%| 62.06%| 99.95%
> > BppAttack|55.83% | 59.84%| 49.01%| 60.18%| 99.33%
> >
> > Guo et al., TABOR: A Highly Accurate Approach to Inspecting and
> > Restoring Trojan Backdoors in AI Systems. ICDM 2020.
> >
> > Hu et al., Trigger Hunting with a Topological Prior for Trojan Detection. ICLR 2022.

---

> > > ### Author Response · Authors · 2022-11-12
> > > **Response to Reviewer 8Hde - Part 3**
> > >
> > > **Q8:** While the experimental results show that the reverse-engineered triggers
> > > can have a high attack success rate, they do not tell us the other side of the
> > > story: false positives. It is common practice to show whether the trigger
> > > inversion helps detect backdoor models and report the detection accuracy among
> > > models. In other words, given a set of clean and poisoned models, how many
> > > instances could the proposed method identify the trigger correctly? How many
> > > clean cases still were identified as malicious?
> > >
> > > **A8:** Thanks for your constructive comment. In our method, we determine a
> > > model is infected with a backdoor if the ASR-Inv of a label is larger than 90%.
> > > The detailed true positive, false positive, false negative, and true negative are
> > > also reported. The dataset and the model used are CIFAR-10 and ResNet18. We
> > > conduct the experiments on attacks in different spaces (i.e., Patch, 1977 Filter
> > > , and WaNet). For each attack, we train 20 benign models and 20 backdoored models.
> > > The results demonstrate that our method achieves high detection accuracy
> > > under the attacks in different spaces. However, existing methods can only handle
> > > the attack in pixel space, and they have low detection accuracy for the attacks
> > > in other spaces.
> > >
> > > Detection Results on Pixel Space Attack (Patch):
> > >
> > > Method | TP| FP | FN | TN | Acc
> > > ---- | ---|--- | --- |--- | ---
> > > NC | 19 | 1 | 1 | 19 | 95.0%
> > > K-arm | 20 | 0 | 0 | 20 | 100.0%
> > > TABOR | 20 | 3 | 0 | 17 | 92.5%
> > > Hu et al. | 18 | 1 | 2 | 19 | 92.5%
> > > UNICORN | 19 | 1 | 1 | 19 | 95.0%
> > >
> > > Detection Results on Signal Space Attack (1977):
> > >
> > > Method | TP| FP | FN | TN | Acc
> > > ---- | ---|--- | --- |--- | ---
> > > NC | 2 | 1 | 18 | 19 | 52.5%
> > > K-arm | 0 | 0 | 20 | 20 | 50.0%
> > > TABOR | 5 | 3 | 15 | 17 | 55.0%
> > > Hu et al. | 4 | 1 | 16 | 19 | 57.5%
> > > UNICORN | 20 | 1 | 0 | 19 | 97.5%
> > >
> > > Detection Results on Feature Space Attack (WaNet):
> > >
> > > Method | TP| FP | FN | TN | Acc
> > > ---- | ---|--- | --- |--- | ---
> > > NC | 8 | 1 | 12 | 19 | 67.5%
> > > K-arm | 9 | 0 | 11 | 20 | 72.5%
> > > TABOR | 3 | 3 | 17 | 17 | 50.0%
> > > Hu et al. | 9 | 1 | 11 | 19 | 70.0%
> > > UNICORN | 18 | 1 | 2 | 19 | 92.5%
> > >
> > > **Q9:** Since the model uses a trainable neural network for trigger inversion,
> > > it would be beneficial to see how it compares to existing methods (such as
> > > Neural Cleanse) in terms of the detection time and computational complexity.
> > >
> > > **A9:** Thanks for your useful comments. We compared the runtime of our method
> > > and that of Neural Cleanse on CIFAR-10 and ResNet18. The result shows our runtime
> > > is 2.7 times of Neural Cleanse's runtime. We admit the computational complexity
> > > of our method is larger than existing methods. However, our method is more
> > > general and robust for inverting different types of backdoor triggers, while
> > > existing methods can only handle pixel space triggers. In addition, our method
> > > can be accelerated by existing works K-arm scheduler (Shen et al.) and mixed
> > > precision training (Micikevicius et al.).
> > >
> > > Shen et al., Backdoor Scanning for Deep Neural Networks through K-Arm Optimization. ICML 2021.
> > >
> > > Micikevicius et al., Mixed Precision Training. ICLR 2018.
> > >
> > > **Q10:** It is unclear what benchmark and dataset were used for the "effects of
> > > the mask size constraint."
> > >
> > > **A10:** Thanks for your valuable comment. The attack, dataset, and model
> > > used in Table 3 are WaNet, CIFAR-10, and EfficientNetB0, respectively. We will
> > > clarify this in the revised version.
> > >
> > > **Q11:** The results in Table 3 need to be repeated for different triggers,
> > > especially those with large sizes.
> > >
> > >
> > > **A11:** Thanks for your helpful suggestion. The results on different triggers
> > > can be found in the following table. The dataset and the model used is CIFAR-10
> > > and EfficientNetB0, respectively. Similar to Table 2, we show the cosine
> > > similarity (SIM) between model's intermediate representations produced by
> > > injected triggers (ground-truth) and the inverted triggers under different
> > > hyper-parameters. A higher SIM value means the intermediate representations of
> > > the inverted trigger are closer to that of the injected trigger. In most cases,
> > > the similarity between the inverted trigger features and the injected trigger
> > > features achieves the highest value when the value of $\beta$ is around 10% of
> > > the entire input space.
> > >
> > > Attack | $\beta$ =1%| $\beta$ =3% | $\beta$ =5% | $\beta$ =10% | $\beta$ =20%
> > > ---- | ---|--- | --- |--- | ---
> > > Patch 3*3 | 0.46 | 0.65 | 0.83 | 0.88 | 0.80
> > > Patch 6*6 | 0.46 | 0.62 | 0.78 | 0.82 | 0.81
> > > Patch 9*9 | 0.44 | 0.55 | 0.69 | 0.77 | 0.79
> > > 1977 | 0.38 | 0.60 | 0.69 | 0.72 | 0.59
> > > WaNet | 0.54 | 0.62 | 0.76 | 0.80 | 0.65
> > > BppAttack | 0.42 | 0.56 | 0.65 | 0.69 | 0.58

---

> > > > ### Author Response · Authors · 2022-11-12
> > > > **Response to Reviewer 8Hde - Part 4**
> > > >
> > > > **Q12:** Given the importance of the disentanglement constraint, what would
> > > > happen if one adds this regularizer to the existing methods, such as Neural
> > > > Cleanse? Can this be used to boost their performance? If so, what is the role of
> > > > the invertible transformation given in Eq. (1)?
> > > >
> > > > **A12:** Thanks for your valuable comment. We conduct the experiment of adding
> > > > the disentanglement constraint on Neural Cleanse (NC). The attack, dataset, and
> > > > model used is 1977, CIFAR-10, and ResNet18, respectively. The results are shown
> > > > in the following table. It demonstrates that the ASR-Inv and the detection
> > > > accuracy are still low when adding the disentanglement constraint on NC. This is
> > > > understandable because the limitation of the existing formulation in the
> > > > inversion problem is that it can not handle the trigger not injected in the
> > > > pixel space. Adding disentanglement constraint on it can not solve the
> > > > limitation. However, the invertible transformation given in Eq. (1) makes
> > > > inverting the trigger in different spaces possible. We will clarify it in the
> > > > revised version.
> > > >
> > > > Method | ASR-Inv| Detection Acc|
> > > > ---- | ---| ---|
> > > > NC | 63.63% | 52.5%|
> > > > NC + Disentanglement Constraint|70.35% | 55.0%|
> > > >
> > > > **Q13:** In Table 5, how is the reverse-engineered trigger's success rate better
> > > > than the original one?
> > > >
> > > > **A13:** Thanks for your helpful question. Neural Cleanse (Wang et al.)
> > > > also has a similar finding that the ASR of the inverted trigger is higher than the
> > > > original one. As pointed out by Neural Cleanse, it is not surprising given the
> > > > trigger is inverted by using a scheme that optimizes for misclassification.
> > > >
> > > > Wang et al., Neural Cleanse: Identifying and Mitigating
> > > > Backdoor Attacks in Neural Networks. IEEE S&P 2019.
> > > >
> > > >
> > > > **Q14:** The effects of different hyper-parameters and how they are set seems
> > > > missing.
> > > >
> > > > **A14:** Thanks for your valuable comments. Besides the influence of $\beta$
> > > > that we already show in A11, the results of the influence of other
> > > > hyper-parameters can be found in the following tables. The dataset and the model
> > > > used is CIFAR-10 and EfficientNetB0. We also show the cosine similarity (SIM)
> > > > between model's intermediate representations produced by injected triggers
> > > > (ground-truth) and the inverted triggers under different hyper-parameters. As
> > > > can be observed, the similarity between the inverted trigger features and the
> > > > injected trigger features is stable when $\alpha<0.01$, $0.85<\gamma<0.95$, and
> > > > $\delta<1.0$. These results show the stability of UNICORN.
> > > >
> > > > Attack | $\alpha =0.005$| $\alpha =0.01$ | $\alpha =0.05$ | $\alpha =0.1$ |
> > > > ---- | ---| --- | --- |--- |
> > > > Patch | 0.87 | 0.88 | 0.82 | 0.80 |
> > > > 1977 | 0.72 | 0.72 | 0.70 | 0.68 |
> > > > WaNet | 0.81 | 0.80 | 0.77 | 0.74 |
> > > > BppAttack | 0.67 | 0.69 | 0.66 | 0.62 |
> > > >
> > > >
> > > > Attack | $\gamma =0.65$| $\gamma =0.75$ | $\gamma =0.85$ | $\gamma =0.90$ | $\gamma =0.95$ |
> > > > ---- | ---|--- | --- |--- |--- |
> > > > Patch | 0.77 | 0.83 | 0.88 | 0.87 |0.79 |
> > > > 1977 | 0.66 | 0.69 | 0.72 | 0.73 | 0.60 |
> > > > WaNet | 0.67 | 0.72 | 0.80 | 0.80 | 0.70 |
> > > > BppAttack | 0.62 | 0.65 | 0.69 | 0.67 | 0.61 |
> > > >
> > > >
> > > > Attack | $\delta =0.5$ | $\delta =1.0$ | $\delta =5.0$ | $\delta =10.0$ |
> > > > ---- | --- | --- |--- | --- |
> > > > Patch |  0.88 | 0.87 | 0.62 | 0.56 |
> > > > 1977 |  0.72 | 0.73 | 0.50 | 0.47 |
> > > > WaNet |  0.80 | 0.81 | 0.64 | 0.55 |
> > > > BppAttack |  0.69 | 0.67 | 0.49 | 0.43 |

---

> ### Author Response · Authors · 2022-11-25
> **A Friendly Reminder**
>
> Dear Reviewer 8Hde,
>
>
> Thanks again for your valuable comments and precious time. We genuinely hope you could have a look at the new results and clarifications and kindly let us know if they have addressed your concerns. We would appreciate the opportunity to engage further if needed.
>
> Best,
>
> Authors of Paper 4378

---

### Official Review · Reviewer_z7uV · 2022-10-23

**Confidence:** 3
**Correctness:** 3
**Technical Novelty And Significance:** 3
**Empirical Novelty And Significance:** 2
**Recommendation:** 6

**Clarity, Quality, Novelty And Reproducibility:**

Clarity: This paper is written clearly.
Quality: The quality is good.
Novelty: The problem and the approach are both novel.
Reproducibility: The authors have provided the code in an anonymous link.


**Strength And Weaknesses:**

Strengths:
1. This is well-written. It tries to propose a uniform framework for backdoor attack and it is novel in that respect.
2. The proposed optimization framework is general and can be applied to various forms of backdoor.
3. The authors have done extensive experiments to validate the effectiveness of the proposed approach.


Weaknesses:
1. I don’t quite follow section 3.3. Many notations, e.g. u and A is not well defined. For now there are only text descriptions. Please formally define these vectors in mathematic equation and write out the dimensions of these vectors. Also many equations are put inline so it becomes even harder to read.
On another note, you were saying the backdoor behaviors in intermediate representations in this part. Is there any empirical evidence to back this claim or is it proposed and established in previous work (if so, please cite the source)? I don’t see why this is obvious without any supporting evidence.
2. The optimization problem in equation 3 involves many parameters to be optimized. Is it stable? I don’t feel a small paragraph in A.3 would convince me that this optimization problem is easy to implement and run in practice. Some plot on the training loss should help readers understand this optimization procedure better. Also how long does this procedure takes compared to previous approaches given that you are optimizing so many parameters at the same time?
3. Why would the formulation in definition 1 generalizes all those previous backdoors you listed? For example, you mentioned wrapping based backdoors, i don’t see what function of phi, m and t would recover such backdoor.
4. How do you decide this target label in equation 1? If you are running equation 1 for every class in the classifier, then how do you decide if a class is poisoned or not?
5. In Figure 3, the inverse trigger does not look similar with the original trigger in some cases, e.g. SIG/Filter/WaNet/BppAttack. Some discussions on this is needed.
6. There are some typos in equation 1. For the first equation, I think M is not needed.



**Summary Of The Paper:**

This paper proposes a new framework for inverting backdoor triggers from backdoor classifiers. Previous approaches is only applicable to patch-based backdoor. The authors proposed an unified approaches. Experiments on various different attacks demonstrate that the proposed approach is able to generate triggers with high attack success rate.

**Summary Of The Review:**

I think this paper is interesting and the problem it tries to tackle is an important one. I am leaning towards accept. But i hope the authors can address the concerns i listed above.

---

> ### Author Response · Authors · 2022-11-12
> **Response to Reviewer z7uV - Part 1**
>
> Thank you very much for your thoughtful comments and recognition of the novelty
> and significance of our work. We hope the following results and clarifications
> can address your concerns.
>
> **Q1:** I don't quite follow section 3.3. Many notations, e.g. $u$ and $A$ is
> not well defined. For now, there are only text descriptions. Please formally
> define these vectors in mathematic equation and write out the dimensions of
> these vectors. We will make it more clear in the revised version.
>
> **A1:** Thank you for your thoughtful comments. The definition of $A$, $u$, and
> $u(x)$ can be found in the following sentences. We will make it more clear in
> the revised version.
>
> * Given a neural network $\mathcal{M}$, we have $h$ and $g$ are the sub-model
> from the input layer to the intermediate layer and the sub-model from the
> intermediate layer to the output layer, respectively. Intuitively, a neural
> network can be split into two functions. There are different ways to split the
> model. In this paper, we split the model at the last convolutional layer.
>
> * Given a neural network $\mathcal{M} = g \circ h$, we
> define the activation vector $A$ as the output of function $h$, i.e., $A = h(x)$,
> where $x$ is the input of the model.
>
> * Given a neural network $\mathcal{M} = g \circ h$, we define $u$ as the unit
> vector (i.e., $|u| = 1$) in the intermediate representation space $N$ mapped by
> function $h$, where $I \xrightarrow{h} N$, $I$ is the input space of the model.
> The dimensions of $u$ are identical to $h(x)$, where $x$ is the input of the
> model.
>
> * Given a neural network $\mathcal{M} = g \circ h$, we
> define $u(x)$ as the projection length on unit vector $u$ for the intermediate
> representation of input $x$, i.e., $u(x) = \frac{h(x) \cdot u} {|u|}$. The projection length $u(x)$ is a scalar.
>
> **Q2:** You were saying the backdoor behaviors in intermediate representations
> in this part. Is there any empirical evidence to back this claim or is it
> proposed and established in previous work (if so, please cite the source)? I
> don't see why this is obvious without any supporting evidence.
>
> **A2:** Thanks for your valuable question. Liu et al. show that some specific
> neurons (compromised neurons) can represent the trigger feature that can
> significantly elevate the probability of the target label when their activation
> value is set to a narrow region, which can support our observation. We also
> conducted experiments on CIFAR-10 and ResNet18 with Patch attack to verify it. We first calculated the compromised
> activation vector $A_c$ and benign activation vector $A_b$ based on their
> definitions. We then perturb $A_b$ by using 100 randomly sampled clean inputs.
> After that, we feed the perturbed $A_b$ and calculate $A_c$ into the submodel
> $g$, and compute the attack success rate, i,e., $P(g(A_c, A_b)=y_t)$, where $y_t$ is the target label. The
> results show that the attack success rate is 95.32% even though the benign
> activation values $A_b$ is perturbed. The results demonstrate that perturbing $A_b$
> will not influence the backdoor behaviors of the infected model, namely, $A_c$
> and $A_b$ are disentangled. We will add more discussion in the revised version.
>
> Liu et al., ABS: Scanning Neural Networks for Back-doors by Artificial
> Brain Stimulation. CCS 2019.
>
> **Q3:** Many equations in section 3.3 are put inline so it becomes even harder
> to read.
>
> **A3:** Thanks for your helpful comment. We will revise accordingly to make it
> more clear.

---

> > ### Author Response · Authors · 2022-11-12
> > **Response to Reviewer z7uV - Part 2**
> >
> > **Q4:** The optimization problem in equation 3 involves many parameters to be
> > optimized. Is it stable? I don't feel a small paragraph in A.3 would convince me
> > that this optimization problem is easy to implement and run in practice. Some
> > plots on the training loss should help readers understand this optimization
> > procedure better.
> >
> > **A4:** Thanks for your valuable comments and suggestions. Our optimization is
> > stable, and it can be supported by existing works. The optimization processes
> > for existing backdoor attacks are also complicated. Many existing backdoor
> > attacks (e.g., Cheng et al., Doan et al. a, Doan et al. b) use generative models as
> > backdoor triggers. In these attacks, the backdoors are injected by
> > simultaneously optimizing the generative networks and the victim models. For
> > example, Cheng et al. optimize the victim model and a complex CycleGAN (Zhu et
> > al.) at the same time to inject the backdoor. Such processes also have many
> > parameters to optimize. Their training loss can reduce smoothly, and they can
> > get high ASR and benign accuracy. These results give us more confidence to
> > invert backdoor triggers via optimizing generative models. We will make it more
> > clear and add plots on the training loss in the revised version.
> >
> > Cheng et al., Deep Feature Space Trojan Attack of Neural Networks by Controlled
> > Detoxification. AAAI 2021.
> >
> > Doan et al. a, Backdoor Attack with Imperceptible Input and Latent Modification.
> > NeurIPS 2021.
> >
> > Doan et al. b, LIRA: Learnable, Imperceptible and Robust Backdoor Attacks. ICCV
> > 2021.
> >
> > Zhu et al., Unpaired Image-to-Image Translation using Cycle-Consistent
> > Adversarial Networks. ICCV 2017
> >
> > **Q5:** How long does this procedure take compared to previous
> > approaches, given that you are optimizing so many parameters at the same time?
> >
> > **A5:** Thanks for your insightful question. We compared the runtime of our
> > method and that of Neural Cleanse on CIFAR-10 and ResNet18. The result shows that our
> > runtime is 2.7 times of Neural Cleanse's runtime. We admit the computational
> > complexity of our method is larger than existing methods. However, our method is
> > more general and robust for inverting different types of backdoor triggers,
> > while existing methods can only handle pixel space triggers. In addition, our
> > method can be accelerated by existing works K-arm scheduler (Shen et al.) and
> > mixed precision training (Micikevicius et al.).
> >
> > Shen et al., Backdoor Scanning for Deep Neural Networks through K-Arm Optimization. ICML 2021.
> >
> > Micikevicius et al., Mixed Precision Training. ICLR 2018.
> >
> > **Q6:** Why would the formulation in definition 1 generalize all those previous
> > backdoors you listed? For example, you mentioned wrapping based backdoors, I
> > don't see what function of $\phi$, $m$ and $t$ would recover such backdoor.
> >
> > **A6:** Thanks for your thoughtful question.
> > * $\phi$ is a function that maps from
> > the pixel space to a specific input space where some elements in it correspond to the
> > backdoor-related content.
> >
> > * For example, in wrapping based backdoors, $\phi$
> > introduce an input space where some elements determine the wrapping strength of
> > the image. $m$ and $t$ correspond to the position of the wrapping effect related
> > elements, and the detailed wrapping strength, respectively.
> >
> > * All existing triggers can be
> > viewed as a particular content in a specific input space. Thus, our definition
> > of backdoor is general.
> >
> > **Q7:** How do you decide this target label in equation 1? If you are running
> > equation 1 for every class in the classifier, then how do you decide if a class
> > is poisoned or not?
> >
> > **A7:** Thanks for your insightful question. Following existing work (Wang et
> > al.), we run it for every class. A class is determined as backdoored if the
> > ASR-Inv for it is larger than 90%.
> >
> > Wang et al., Neural Cleanse: Identifying and Mitigating
> > Backdoor Attacks in Neural Networks. IEEE S&P 2019.
> >
> > **Q8:** In Figure 3, the inverse trigger does not look similar with the original
> > trigger in some cases, e.g., SIG/Filter/WaNet/BppAttack. Some discussions on this
> > are needed.
> >
> > **A8:** Thanks for your valuable question. It is because these backdoor attacks
> > are inaccurate (Wang et al., Nguyen et al.). In other words, a trigger that is
> > not identical to the injected one can also activate the model's backdoor
> > behaviors. We observed that pixel space attacks are more accurate than the
> > attacks in other spaces. We will make it more clear in the revised version.
> >
> > Wang et al., Neural Network Trojans Analysis and Mitigation from the Input Domain. arXiv 2022.
> >
> > Nguyen et al., WaNet -- Imperceptible Warping-based Backdoor Attack. ICLR 2021.
> >
> >
> > **Q9:** There are some typos in equation 1. For the first equation, I think
> > $\mathcal{M}$ is not needed.
> >
> > **A9:** Thanks for your helpful comment. In equation 1, $\mathcal{M}$ means the
> > suspicious model, and $\mathcal{M}(\tilde{x})$ is the model's output on
> > the samples pasted with the inverted trigger. We will make it more clear. We will
> > carefully check typos.

---

### Official Review · Reviewer_kjQZ · 2022-10-25

**Confidence:** 3
**Correctness:** 3
**Technical Novelty And Significance:** 3
**Empirical Novelty And Significance:** 3
**Recommendation:** 6

**Clarity, Quality, Novelty And Reproducibility:**

- Clarity can be improved. (-)
- The quality of this paper is good. (+)
- Novelty is strong. (++)
- Code is provided (reproducibility).  (++)

**Details Of Ethics Concerns:**

No ethics concerns.

**Strength And Weaknesses:**

B. Strength
- The visualization is good.
- The novelties and motivations are strong.

C. Weakness
- Some of the design details are missing.
- The paper presentation can be improved.
- The results are not easy to interpret.

**Summary Of The Paper:**

A. Paper summary:

- This paper proposes a trigger inversion framework that can be generalized to different types of triggers. Specifically, this paper introduces a new optimization problem on top of the previous trigger detection formulation with an input transformation and inversion function. Using two neural networks to approximate the transformation and inversion functions, the method can be generalized to all types of triggers. The result shows strong performance in reverse engineering trojan patterns.

**Summary Of The Review:**

D. Questions:
- ASR-Inj is applied in both Table 4 and Table 5 and it is introduced at the end of the evaluation function. Maybe you should reorganize the evaluation section a little bit.

- You use ASR-Inv as the key metric throughout the paper. So I am confused by the goal of this paper. Aren't you trying to detect if a model is poisoned or not? What is the prediction accuracy of testing trojan models using your proposed method?

- Also, you never mentioned how you implement P and Q. How you designed the neural networks to fulfill the inversion and transformation?

- A lot of ablation studies can be added. For example, the portion of the training dataset that is required to reverse-engineer the trojan.


Overall, the novelty of this paper is strong.

---

> ### Author Response · Authors · 2022-11-12
> **Response to Reviewer kjQZ - Part 1**
>
> Thank you very much for your valuable comments and recognition of the novelty. We hope the following results and clarifications can address your
> concerns.
>
> **Q1:** ASR-Inj is applied in both Table 4 and Table 5 and it is introduced at
> the end of the evaluation function. Maybe you should reorganize the evaluation
> section a little bit.
>
> **A1:** Thanks for your valuable comment. We will reorganize the evaluation
> section and add the definition of the ASR-Inj at the beginning of the section.
>
> **Q2:** You use ASR-Inv as the key metric throughout the paper. So I am confused
> by the goal of this paper.
>
> **A2:** Thanks for your constructive comment.
>
> * Our goal is to invert the backdoor trigger in the models. Trigger inversion
> methods can provide key information about the attacks (e.g., the target label
> and the shape of the trigger) and help the user to understand/mitigate the
> backdoor attacks. Following existing work (Tao et al.), we use the ASR (attack
> success rate) of the inverted trigger to measure if the inversion is successful
> because high ASR is the fundamental goal of the backdoor attack.
>
> * We also report the results under different metrics in the rebuttal, such as
> backdoored models detection accuracy (A3) and the cosine similarity between
> model's intermediate representations produced by injected triggers
> (ground-truth) and the inverted triggers (A5).
>
> Tao et al., Better Trigger Inversion Optimization in Backdoor Scanning. CVPR 2022.
>
> **Q3:** What is the prediction accuracy of testing trojan models using your
> proposed method?
>
> **A3:** Thanks for your insightful question. In our method, we determine a model
> is infected with a backdoor if the ASR-Inv of a label is larger than 90%. The
> detailed true positive, false positive, false negative, and true negative are also
> reported. The dataset and the model used are CIFAR-10 and ResNet18. We conduct
> experiments on attacks in different spaces (i.e., Patch, 1977 Filter, and
> WaNet). For each attack, we train 20 benign models and 20 backdoored models. The
> results demonstrate that our method achieves high detection accuracy under
> attacks in different spaces. However, existing methods can only handle the
> attack in pixel space and have low detection accuracy for the attacks in
> other spaces.
>
> Detection Results on Pixel Space Attack (Patch):
>
> Method | TP| FP | FN | TN | Acc
> ---- | ---|--- | --- |--- | ---
> NC | 19 | 1 | 1 | 19 | 95.0%
> K-arm | 20 | 0 | 0 | 20 | 100.0%
> TABOR | 20 | 3 | 0 | 17 | 92.5%
> Hu et al. | 18 | 1 | 2 | 19 | 92.5%
> UNICORN | 19 | 1 | 1 | 19 | 95.0%
>
> Detection Results on Signal Space Attack (1977):
>
> Method | TP| FP | FN | TN | Acc
> ---- | ---|--- | --- |--- | ---
> NC | 2 | 1 | 18 | 19 | 52.5%
> K-arm | 0 | 0 | 20 | 20 | 50.0%
> TABOR | 5 | 3 | 15 | 17 | 55.0%
> Hu et al. | 4 | 1 | 16 | 19 | 57.5%
> UNICORN | 20 | 1 | 0 | 19 | 97.5%
>
> Detection Results on Feature Space Attack (WaNet):
>
> Method | TP| FP | FN | TN | Acc
> ---- | ---|--- | --- |--- | ---
> NC | 8 | 1 | 12 | 19 | 67.5%
> K-arm | 9 | 0 | 11 | 20 | 72.5%
> TABOR | 3 | 3 | 17 | 17 | 50.0%
> Hu et al. | 9 | 1 | 11 | 19 | 70.0%
> UNICORN | 18 | 1 | 2 | 19 | 92.5%
>
> **Q4:** Also, you never mentioned how you implement P and Q. How you designed
> the neural networks to fulfill the inversion and transformation?
>
> **A4:** Thanks for your helpful comment. We use representative deep neural
> network UNet (Ronneberger et al.) to model the space transformation functions
> because of its high expressiveness. P and Q are represented by two identical UNet
> networks. UNet is widely used in image transformation tasks, such as image style
> transformation and image attribute editing. We will make it more clear in the
> revised version.
>
> Ronneberger et al., Convolutional Networks for Biomedical Image Segmentation.
> MICCAI 2015

---

> > ### Author Response · Authors · 2022-11-12
> > **Response to Reviewer kjQZ - Part 2**
> >
> > **Q5:** A lot of ablation studies can be added. For example, the portion of the
> > training dataset that is required to reverse-engineer the trojan.
> >
> > **A5:** Thanks for your constructive suggestion.
> >
> > * The results under different portions $p$ of the required training dataset can
> > be found in the following table. The dataset and the model used is CIFAR-10 and
> > EfficientNetB0. Similar to Table 2, we show the cosine similarity (SIM) between
> > model's intermediate representations produced by injected triggers
> > (ground-truth) and the inverted triggers under different hyper-parameters. A
> > higher SIM value means the intermediate representation of the inverted trigger
> > is closer to that of the injected trigger. As can be observed, the SIM value is
> > low when the portion is smaller than 0.10% (5 samples per class). When it is
> > larger than 0.20% (10 samples per class), the results are stable, and UNICORN
> > achieves high similarity between the inverted trigger features and the injected
> > trigger features. We will add more results in the revised version.
> >
> > Attack | $p$ = 0.04%| $p$ = 0.10% | $p$ = 0.20% | $p$ = 1.00% |
> > ---- | ---|--- | --- |--- |
> > Patch | 0.70 | 0.88 | 0.88 | 0.89 |
> > 1977 | 0.61 | 0.66 | 0.72 | 0.72 |
> > WaNet | 0.74 | 0.78 | 0.80 | 0.81 |
> > BppAttack | 0.56 | 0.60 | 0.69 | 0.68 |
> >
> > * We also conducted other ablation studies. Besides the influence of the portion
> > of the required training dataset and hyper-parameter $\beta$ that we already
> > show in Table 2, the results of the influence of other hyper-parameters can be
> > found in the following tables. The dataset and the model used is CIFAR-10 and
> > EfficientNetB0. We show the cosine similarity (SIM) between model's intermediate
> > representations produced by injected triggers (ground-truth) and the inverted
> > triggers under different hyper-parameters. As can be observed, the similarity
> > between the inverted trigger features and the injected trigger features is
> > stable when $\alpha<0.01$, $0.85<\gamma<0.95$, and $\delta<1.0$. These results
> > show the stability of UNICORN. We are happy to conduct more experiments if
> > needed.
> >
> > Attack | $\alpha =0.005$| $\alpha =0.01$ | $\alpha =0.05$ | $\alpha =0.1$ |
> > ---- | ---| --- | --- |--- |
> > Patch | 0.87 | 0.88 | 0.82 | 0.80 |
> > 1977 | 0.72 | 0.72 | 0.70 | 0.68 |
> > WaNet | 0.81 | 0.80 | 0.77 | 0.74 |
> > BppAttack | 0.67 | 0.69 | 0.66 | 0.62 |
> >
> >
> > Attack | $\gamma =0.65$| $\gamma =0.75$ | $\gamma =0.85$ | $\gamma =0.90$ | $\gamma =0.95$ |
> > ---- | ---|--- | --- |--- |--- |
> > Patch | 0.77 | 0.83 | 0.88 | 0.87 |0.79 |
> > 1977 | 0.66 | 0.69 | 0.72 | 0.73 | 0.60 |
> > WaNet | 0.67 | 0.72 | 0.80 | 0.80 | 0.70 |
> > BppAttack | 0.62 | 0.65 | 0.69 | 0.67 | 0.61 |
> >
> >
> > Attack | $\delta =0.5$ | $\delta =1.0$ | $\delta =5.0$ | $\delta =10.0$ |
> > ---- | --- | --- |--- | --- |
> > Patch |  0.88 | 0.87 | 0.62 | 0.56 |
> > 1977 |  0.72 | 0.73 | 0.50 | 0.47 |
> > WaNet |  0.80 | 0.81 | 0.64 | 0.55 |
> > BppAttack |  0.69 | 0.67 | 0.49 | 0.43 |

---

### Author Response · Authors · 2022-11-18
**Revision Summary**

We sincerely thank all reviewers for their thoughtful comments and precious
time. We are glad that all reviewers found our paper "novel and
interesting". We provide our responses to address the concerns. Our paper has
been revised accordingly. Below is our revision summary:


**[Abstract, Section 1]** We revised the sentences mentioning the limitation of
existing trigger inversion methods, following the suggestion of Reviewer 8Hde.

**[Section 2]** We added more discussion about the motivation behind trigger
inversion methods, following the suggestion of Reviewer 8Hde.

**[Section 3]** We added clarification for the motivation of the material
presented in Section 3.3, following the suggestion of Reviewer 8Hde. We revised
the format of the equations in Section 3.3 to increase readability, following
the suggestion of Reviewer z7uV.

**[Section 3, Appendix Section A.3]** We added the discussion about existing
works related to the backdoor model's inner behaviors, and clarified the reason
why we use the proposed constraint among different regularizers, following the
suggestion of Reviewer 8Hde.

**[Section 4]** We added clarification for the implementation of P and Q,
following the suggestion of Reviewer kjQZ. We reorganized Section 4 by first reasoning about different properties of the inverted trigger and then presenting the mathematical expressions, following the suggestion of Reviewer 8Hde.

**[Section 5]** We added clarification and discussion for the used metrics,
following the suggestion of Reviewer kjQZ. We added the definition of the
ASR-Inj at the beginning of the section, following the suggestion of Reviewer
kjQZ. We added more discussion about the cases where inverted trigger
does not look similar to the injected trigger, following the suggestion of
Reviewer z7uV. We added the explanation about the cases where the ASR of the
inverted trigger is higher than that of the injected trigger, following the
suggestion of Reviewer 8Hde.

**[Section 5, Appendix Section A.7]** We added the results on more baselines, following the suggestion of Reviewer 8Hde.

**[Appendix Section A.1]** We added formal definitions of the notations used in
Section 3, following the suggestion of Reviewer z7uV.

**[Appendix Section A.2]** We added empirical results and discussions to support
Corollary 1, following the suggestion of Reviewer z7uV.

**[Appendix Section A.7]** We added the backdoor models detection results, following
the suggestion of Reviewer kjQZ and Reviewer 8Hde.

**[Appendix Section A.8]** We added more ablation studies to investigate the
influence of hyper-parameters, following the suggestion of Reviewer kjQZ and
Reviewer 8Hde. We added clarification for the model and dataset used in the
experiments investigating the influence of the mask size constraint, following
the suggestion of Reviewer 8Hde. We added the results for larger patch triggers,
following the suggestion of Reviewer 8Hde.

**[Appendix Section A.9]** We added the discussion about the efficiency of our
method, following the suggestion of Reviewer z7uV and Reviewer 8Hde.

**[Appendix Section A.10]** We added the plots of training loss and the
discussion about the stability of our optimization process, following the
suggestion of Reviewer z7uV.

**[Appendix Section A.11]** We added the results and discussion for adding the
disentanglement constraint to existing methods, following the suggestion of
Reviewer 8Hde.

Please let us know if there is anything still not clear. We are willing to
answer more questions and perform more experiments if the reviewers have further
concerns. We really look forward to the discussion with the reviewers to further
improve our paper.

---

### Decision · Program_Chairs · 2023-01-20

**Decision:**

Accept: notable-top-25%

**Justification For Why Not Higher Score:**

N/A

**Justification For Why Not Lower Score:**

The work presents significant and novel insights on a very relevant and technical challenging problem.

**Metareview: Summary, Strengths And Weaknesses:**

This paper provides a formal definition and analysis of triggers in the context of a backdoor attack on Deep Neural Network (DNN) models. It then proposes a unified framework to invert backdoor triggers based on the formalization of triggers and the identified inner behaviors of backdoor models. Finally, the authors provide a prototype called UNICORN which is general and effective in inverting backdoor triggers in DNNs.

Strengths:
- Well-written paper proposing a uniform framework for backdoor attack, Visualization is good
- General optimization framework can be applied to various forms of backdoor
- Extensive experiments to validate effectiveness of proposed approach
- Code is provided (reproducibility)
- A broad range of backdoor attacks and neural network architectures are used in the experimental results.

Weaknesses:
- Paper presentation and clarity can be improved (eg Motivations behind the proposed approach during the introduction/Section 3.3 not well defined and notations not formally defined/Section 4)
- Related to point above: Some of the design details are missing and sometimes the results were not easy to interpret
- Questions still remain on implementation of P and Q, prediction accuracy, and ablation studies
- No empirical evidence to back claim of backdoor behaviors in intermediate representations
- Optimization problem involves many parameters (and is not well explained, related to point 1 above)
- Not clear how target label in equation 1 is decided
- Inverse trigger does not look similar to original trigger in some cases
- The number of baselines used in the paper is limited, and experimental results do not highlight cases of false positives.
-The computational complexity of the proposed methods need to be compared against baselines methods.


Although the reviewers noted many points of concern the rebuttal was very thorough from the authors perspective, and addressed many of them. The authors were quite extensive in providing many points of clarification (which was in fact the biggest concern of all reviewers). Some reviewer engagement was present post rebuttal, and all converged (including myself) on a positive assessment of the novelty and intellectual contribution of this work.



**Note From Pc:**

if the above contains the word "oral" or "spotlight" please see: "oral" presentation means -> notable-top-5% and "spotlight" means -> notable-top-25%. As stated in our emails, we are disassociating presentation type from AC recommendations

**Summary Of Ac-Reviewer Meeting:**

N/A